# EFFICIENT SELF-SUPERVISED VISION TRANSFORMERS FOR REPRESENTATION LEARNING

**Chunyuan Li**[1]  **Jianwei Yang**[1]  **Pengchuan Zhang**[1]  **Mei Gao**[2]  **Bin Xiao**[2]  **Xiyang Dai**[2]
**Lu Yuan**[2]  **Jianfeng Gao**[1]
[1]Microsoft Research at Redmond, [2]Microsoft Cloud + AI
`{chunyl,jianwyan,penzhan,xuga,bixi,xidai,luyuan,jfgao}@microsoft.com`

## ABSTRACT

This paper investigates two techniques for developing efficient self-supervised vision transformers (EsViT) for visual representation learning. First, we show through a comprehensive empirical study that multi-stage architectures with sparse self-attentions can significantly reduce modeling complexity but with a cost of losing the ability to capture fine-grained correspondences between image regions. Second, we propose a new pre-training task of region matching which allows the model to capture fine-grained region dependencies and as a result significantly improves the quality of the learned vision representations. Our results show that combining the two techniques, EsViT achieves 81.3% top-1 accuracy on the ImageNet linear probe evaluation, outperforming prior arts with around an order magnitude of higher throughput. When transferring to downstream linear classification tasks, EsViT outperforms its supervised counterpart on 17 out of 18 datasets. The code and pre-trained models are released at: `https://github.com/microsoft/esvit`

## 1 INTRODUCTION

Self-supervised learning (SSL) with Transformers (Vaswani et al., 2017) has become a de facto standard of model choice in natural language processing (NLP). The dominant approaches such as GPT (Radford et al., 2018) and BERT (Devlin et al., 2019) are pre-training on a large text corpus and then fine-tuning to various smaller task-specific datasets, showing superior performance. Larger Transformers pre-trained with larger-scale language datasets often lead to a stronger generalization ability, demonstrated by improved performance in downsteam tasks (with no sign of performance saturation yet), as exemplified in GPT-3 (Brown et al., 2020).

In computer vision (CV), however, self-supervised visual representation learning is still dominated by convolutional neural networks (CNNs). Sharing a similar goal/spirit with NLP, SSL in CV aims to learn general-purpose image features from raw pixels without relying on manual supervisions, and the learned networks are expected to serve as the backbone of various downstream tasks such as classification, detection and segmentation. Recently, impressive performance have been achieved by CNN-based SSL, outperforming state-of-the-art (SoTA) fully-supervised pre-training methods (He et al., 2020; Caron et al., 2020) on tasks with a limited number of labels. The key to success is view-level learning: maximizing agreement of learned representations between differently augmented views of the same example. Recent works, including SimCLR-v2 (Chen et al., 2020d), BYOL (Grill et al., 2020) and SwAV (Caron et al., 2020), have scaled up the CNN-based models to hundreds of millions of parameters. However, SSL has not enjoyed the same scaling success in CV as that in NLP.

Several attempts have been made to close the gap by combining SSL with Transformer and self-attention architectures. Early works include Selfie (Trinh et al., 2019), which generalizes the concept of masked language modeling of BERT for images. The idea has been recently revisited in Vision Transformer (ViT) (Dosovitskiy et al., 2021) via pre-training on a much larger scale dataset, *e.g.,* JFT-300M. ImageGPT (iGPT) (Chen et al., 2020b) generalizes the concept of auto-regressive language modeling of GPT for images, showing encouraging ImageNet recognition accuracy with a large model size. Contrastive learning with ViT has also been studied very recently in DINO (Caron et al., 2021) and MoCo-v3 (Chen et al., 2021), where new SoTA result by linear probe evaluation on

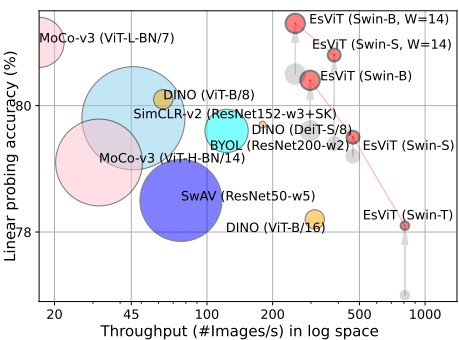 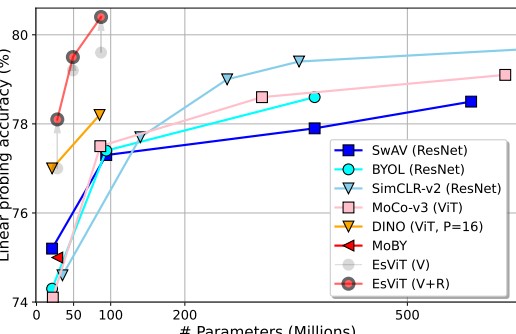

Figure 1: Efficiency vs accuracy comparison under the linear classification protocol on ImageNet. Left: Throughput of all SoTA SSL vision systems, circle sizes indicates model parameter counts; Right: performance over varied parameter counts for models with moderate (throughout/#parameters) ratio. EsViT pre-trained with and without the region-matching task are shown before and after the arrows, respectively. Please refer Section 4.1 for details.

ImageNet-1K is achieved, by exhaustively consuming computation resource on full self-attention operators with long sequences of split image patches.

Aiming to improve the *efficiency* of Transformer-based SSL, this paper presents **E***fficient* **s***elf-superivsed* **Vi***sion* **T***ransformers* (EsViT), by using a multi-stage architecture and a region-based pre-training task for self-supervised representation learning. Our main findings and contributions can be summarized as follows:

(1) An intriguing property of self-supervised monolithic Transformers is firstly reported in our paper: automatic discovery of semantic correspondence between local regions.

(2) We present the first comprehensive empirical study to show the pros and cons of multi-stage vision Transformer architectures for SSL. Though greatly reducing compute complexity, we find that the multi-stage architecture causes the loss of the property in (1).

(3) A region matching pre-train task is proposed to alleviate the issue in (2), and further improve the learned representations and attentions.

(4) We validate the new EsViT, which combines the two techniques, on a range of tasks. It significantly reduces the cost in building SoTA SSL vision systems, as summarized in Figure 1, and shows better scaling performance on accuracy vs. throughput and model size. Under the linear evaluation protocol, EsViT achieves 81.3% top-1 accuracy, showing the best performance compared with all systems, and is $3.5\times$ parameter-efficient and has at least $10\times$ higher throughput than previous SoTA (81.0%, MoCo-v3 with ViT-BN-L/7 (Chen et al., 2021)). Compared with its supervised counterpart Swin Transformers (Liu et al., 2021), EsViT shows superior performance on 17 out 18 datasets, when transferring the learned representations to downstream linear classification tasks.

## 2 METHODS

Transformer-based SSL methods emerge very recently to lead the state-of-the-art performance on the ImageNet linear probe task (Chen et al., 2021; Caron et al., 2021). It inherits the successes from (1) monolithic Transformer architectures that dominate in NLP (Devlin et al., 2019; Radford et al., 2018), and (2) instance-level contrastive learning objectives that demonstrate arguably the best SSL performance in computer vision (Chen et al., 2020c). Though simple and effective, the existing Transformer-based SSL methods require a large amount of compute resources (*e.g.,* >1.7 TPU years of training) to reach SoTA performance. We believe that the SSL system *efficiency* is highly related to two ingredients: the network architecture and the pre-train task. To strike for a better tradeoff between accuracy and efficiency, we present EsViT, showing better synergy of networks (a multi-stage Transformer architecture) and pre-train tasks (a non-contrastive region-matching task).

### 2.1 NETWORK ARCHITECTURES: FROM MONOLITHIC TO MULTI-STAGE VIT BACKBONE

**Multi-stage ViT.** This paper presents the first empirical study of multi-stage Transformer architectures (Vaswani et al., 2021; Wang et al., 2021; Liu et al., 2021; Zhang et al., 2021; Wu et al., 2021)

for SSL. Each stage consists of a *patch merging/embedding* module, and a *Transformer with sparse self-attention* module. (*i*) The patch merging module plays a slightly different roles in different stages. In the first stage, it splits an input RGB image into non-overlapping patches. Each patch is treated as a "token", constructed as a concatenation of the raw pixel RGB values, which is further projected into a $C$-dimension feature. In the later stage, the patch merging module concatenates the features of each group of $2 \times 2$ neighboring patches, and applies a linear layer on the $4C$-dimensional concatenated features. This reduces the number of tokens by a multiple of $2 \times 2 = 4$, and the output dimension is set to $2C$. (*ii*) A Transformer with sparse self-attention module are then employed to enable interactions among the merged features. The two modules above are repeated for multiple times, typically 4 times, resulting in a multi-stage ViT. As a result, a hierarchical representation is generated: the number of tokens is reduced and the feature dimension (and the number of heads in self-attentions) of each token is increased, as the network gets deeper. An overview comparison of the monolithic and multi-stage Transformer architectures for SSL is illustrated in Figure 7 in Appendix.

**An intriguing property of self-supervised monolithic ViT.** Though straightforward in implementation, changing from monolithic to multi-stage architecture without careful treatments may lose some desirable properties of self-supervised Transformers In out study, we first empirically note an intriguing property of self-supervised monolithic ViT(Caron et al., 2021): *the pre-trained model exhibits a very strong ability to automatically discovers correspondences, even without a region-level matching objective specified in training*.

We quantitatively evaluate the correspondence learning to illustrate this property, as discussed in the following process. (*i*) *Simulated benchmark*. Based on 50K images in the ImageNet validation dataset, we create a simple evaluation benchmark with mild augmentations: For a center-crop image, we apply `HorizontalFlip`, then `ColorJitter` and `RandomGrayscale` to create a new augmented view. In this way, ground-truth correspondences are created. (*ii*) *Evaluation process*. Given two views of the same image, we use the pre-trained backbone to extract the top-layer features, and find the feature vector in one view that best matches the other in terms of highest cosine similarity. The accuracy is measured as the averaged percentage of correctly identifying the region-to-region correspondences. Please see details in Section C.7 in Appendix. (*iii*) *Results*. We quantitatively show that a self-supervised monolithic ViT yields **95%** accuracy. However, simply replacing the network with a multi-stage Transformer yields only **66%** accuracy. This significant degradation (absolute **29%** accuracy drop) reveals the loss of the correspondence learning property. We first raise this critical problem, and believe that it has a large impact on the pre-trained model's performance in various downstream tasks.

## 2.2 Pre-training Tasks: Delving into Views with Regions

We employ a non-contrastive learning framework to build our SSL method. Specifically, *Self-distillation with no labels* (DINO) (Caron et al., 2021) is considered. It leverages the knowledge distillation learning paradigm where a student network $g_{\theta_s}$ is trained to match the output of a given teacher network $g_{\theta_t}$, parameterized by $\theta_s$ and $\theta_t$ respectively. The neural network $g$ is composed of a backbone $f$ (*e.g.,* Transformers or ConvNets), and of a projection head $h$: $g = h \circ f$. The features used in downstream tasks are the output of backbone $f$. In SSL, different augmented views $\tilde{x}$ of an image $x$ are fed into backbone network to obtain feature maps $z = f(\tilde{x})$. Two MLP heads followed by `softmax` per network further convert the feature vectors $z \in \boldsymbol{z}$ into probability vectors $p = h(z)$; one head for view-level and the other head for region-level, respectively.

More precisely, from a given image, we generate a set $\mathcal{V}$ of different views[1] following (Caron et al., 2021). The resulting feature map at the top layer for each view is $\boldsymbol{z} = [z_1, \ldots, z_T]$, where $T$ is the sequence length, and $z_i$ is a region-level representation for the local patch at position $i$. Average pooling is applied to obtain the view-level representation $\bar{z} = \text{avg-pool}(\boldsymbol{z})$.

**View-level task** Given the augmented view set for student $\mathcal{V}$ and teacher $\mathcal{V}^*$, a set of pairs $\mathcal{P} = \{(s,t)|\tilde{\boldsymbol{x}}_s \in \mathcal{V}, \tilde{\boldsymbol{x}}_t \in \mathcal{V}^* \text{ and } s \neq t \}$ is constructed to perform cross-view prediction tasks. We

---

[1]This set often contains views of two different resolutions $\mathcal{V} = [\mathcal{V}_g, \mathcal{V}_l]$, where $\mathcal{V}_g = \{\tilde{\boldsymbol{x}}_{g_i}|i = 1, 2\}$ is a global-view set of higher resolution, and $\mathcal{V}_l = \{\tilde{\boldsymbol{x}}_{l_i}|i = 1, \ldots, 8\}$ is a local-view set of lower resolution. All views $\mathcal{V}$ are passed through the student while only the global views $\mathcal{V}_g$ are passed through the teacher.

consider the pre-training task at the view level proposed by (Caron et al., 2021):

$$\mathcal{L}_V = \frac{1}{|\mathcal{P}|} \sum_{(s,t)\in\mathcal{P}} \mathcal{M}_V(s,t), \ \ \text{with} \ \ \mathcal{M}_V(s,t) = -p_s \log p_t, \tag{1}$$

where $p_s = h(\bar{z}_s)$ and $p_t = h(\bar{z}_t)$ are the probability output of an MLP head $h$ over the view-level representations $\bar{z}_s$ and $\bar{z}_t$, learned by student and teacher, respectively. In DINO, ViT/DeiT are considered, hence the view-level representation is the feature of the [CLS] token.

**Region-level task**   In (Caron et al., 2021), the $\mathcal{L}_V$ encourages "local-to-global" correspondences only at a coarse level: the large crop and the small crop are matched in the view level, leaving region-to-region correspondence unspecified. In monolithic Transformers, the drop paths and skip connections from low-level features to high-level features help the the latter to remain discriminative, thus maintain good region-matching performance. However, such a property gets diluted due to the merging operators in multi-stage Transformers. As shown in our experiments later, training a multi-stage network with $\mathcal{L}_V$ only indeed results in sub-optimal representations.

Further, it could be a waste of computation not to leverage region-level features $z$ that are computed in the process of extracting view-level feature. Inspired by the success of masked language modeling task in BERT, we argue that it is important to have region-level pre-training task for computer vision, so that the model can (1) amortize the computation and fully leverage the extracted region-level features, and (2) take into account the co-occurrences/structures between local features. Unfortunately, directly performing masked patch prediction (MPP) for the multi-stage Transformer architecture is infeasible, as the one-to-one correspondences between the input visual tokens and output features get diluted due to the merging operation. Even for monolithic architectures, MPP has not been proved effective in computer vision, as empirically shown in (Dosovitskiy et al., 2021).

To address this problem, we propose a non-contrastive, region-matching method that directly works at the level of local features by taking into account their correspondences:

$$\mathcal{L}_R = \frac{1}{|\mathcal{P}|} \sum_{(s,t)\in\mathcal{P}} \mathcal{M}_R(s,t), \ \ \text{with} \ \ \mathcal{M}_R(s,t) = -\frac{1}{T} \sum_{i=1}^{T} p_{j^*} \log p_i, \ \ j^* = \arg\max_j \frac{z_i^T z_j}{\|z_i\|\|z_j\|}, \tag{2}$$

where $p_i = h'(z_i)$ and $p_j = h'(z_j)$ are the probability outputs of a new MLP head $h'$ over the local features of student $z_i \in \boldsymbol{z}_s$ and teacher $z_j \in \boldsymbol{z}_t$, respectively. $j^*$ is the index of the feature in $\boldsymbol{z}_t$ that best matches the $i$-th feature in $\boldsymbol{z}_s$, in the sense of highest cosine similarity. Note that $z_i$ and $z_{j^*}$ are *contextualized* features of two best matched regions from different augmented views, minimizing $\mathcal{L}_R$ encourages different contexts (*i.e.,* surrounding regions) to learn invariant features, and thus captures the region-dependency.

The overall pre-training objective of EsViT is $\mathcal{L} = \mathcal{L}_R + \mathcal{L}_V$, we learn to match the feature distributions at both the view and region levels by minimizing the cross-entropy loss w.r.t. the parameters of the student network $g_{\boldsymbol{\theta}_s}$. A visual illustration is in Figure 2, and the full algorithm is in Appendix. We updates teacher/student network alternatively: (*i*) Given a fixed teacher network, the student network is updated by minimizing the full cross-entropy loss: $\boldsymbol{\theta}_s \leftarrow \arg\min_{\boldsymbol{\theta}_s} \mathcal{L}(s,t;\boldsymbol{\theta}_s)$. (*ii*) The teacher model is updated as an exponential mov-

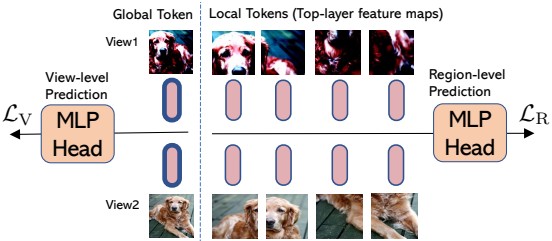

Figure 2: Pre-training objectives, including view-level (left) and region-level (right) prediction.

ing average (EMA) of the student weights $\boldsymbol{\theta}_t \leftarrow \lambda\boldsymbol{\theta}_t + (1-\lambda)\boldsymbol{\theta}_s$, with $\lambda$ following a cosine schedule from 0.996 to 1 during training. By default, the full objective $\mathcal{L}$ is used from the beginning. One can also load a checkpoint trained by $\mathcal{L}_V$ only, and add $\mathcal{L}_R$ for continual pre-training, which is shown effective in boosting performance in our experiments.

**Computational overhead**   Note that applying $\mathcal{L}_R$ on the traditional monolithic Transformer architecture can be prohibitively computationally expensive, as it requires $\mathcal{O}(T^2)$ to compute $\mathcal{L}_R$. For a typical image of resolution $224 \times 224$, the feature map length of ViT/DeiT (with patch size 16) at

the top layer is $T = 196$, while the multi-stage architecture yields $T = 49$, which requires 3 times less compute in computing $\mathcal{L}_R$. To empirically illustrate this, we show in Appendix Section C.2 that $\mathcal{L}_R$ adds acceptable extra memory and computational cost (around 1.2 and 1.05 ×, respectively) for multi-stage Transformers, while it will quickly go out-of-memory for monolithic Transformers when the batch size is increased.

## 3 RELATED WORKS

**Relation to mask prediction tasks**    We can consider the proposed $\mathcal{L}_R$ as a proxy to mimick masked language modeling in BERT, where the "ground-truth" local token is a soft label provided by the teacher network, while the student network makes predictions to match that target, based on the context of regions in a different augmented view. Importantly, our $\mathcal{L}_R$ considers `softmax` with cross-entropy in the objective, rather than `MSE` as in MPP. A very sharp teacher distribution is used by choosing small temperatures. This encourages the model to focus on the salient dimensions, rather than waste modeling capability on training short-range dependencies and high-frequency details (Ramesh et al., 2021).

**Relation to DenseCL**    The proposed $\mathcal{L}_R$ mostly related to DenseCL (Wang et al., 2020b) in that the region correspondences in both methods are determined as the two most similar grid features. One critical difference is that DenseCL is a contrastive region-matching task, while our $\mathcal{L}_R$ is a non-contrastive region-matching task, where no negative samples/queue is needed. This technical difference has a significant impact on the downstream task performance. We find that $\mathcal{L}_R$ is particularly effective in serving our goal to improve image classification performance and build efficient & affordable SoTA SSL system; In contrast, DenseCL degrades the classification performance.

**Relation to other region-level tasks**    The ideas of leveraging local region-level pre-training tasks for visual representation learning have been explored for ConvNets (Misra & Maaten, 2020; Xiong et al., 2020; Wang et al., 2020b; Xie et al., 2021a; Yang et al., 2021; Xie et al., 2021c). We summarize the differences in three aspects: (*i*) Motivation. Our region-matching task $\mathcal{L}_R$ aims to recover the lost property of automatic correspondence learning in self-supervised monolithic Transformers, while most existing region-level tasks aim to improve dense visual prediction tasks. (*ii*) Technical difference. Our $\mathcal{L}_R$ is a non-contrastive region-matching task, while others are contrastive learning. (*iii*) Empirical performance. Most region-level tasks improve dense visual prediction tasks but sacrifice their image classification performance, while $\mathcal{L}_R$ consistently improves classification performance. Among them, EsViT training method achieves the best ImageNet linear probe performance with minimum computational overhead. For detailed comparisons, please refer to Table 7 in Appendix.

**Self-supervised vision Transformers.**    The research on Transformer-based self-supervised representation learning just scratches the tip of the iceberg, and only a few attempts are made on this topic. ImageGPT (Chen et al., 2020b) and MoCo-v3 (Chen et al., 2021) dedicate huge compute resource with large models to exploring the frontier. DINO (Caron et al., 2021) achieves comparable performance of large self-supervised ConvNets using small/medium-size Transformers. The proposed EsViT further pursues efficient and affordable solutions to self-supervised vision Transformers. For more general related works on Transformers for vision tasks and self-supervised ConvNets, please refer to Section B in Appendix.

## 4 EXPERIMENTAL RESULTS

We describe the experimental settings in Appendix Section C.3, and evaluate the proposed EsViT to answer three questions: **Q1**: How does EsViT perform on standard ImageNet benchmark compared to SoTA methods? **Q2**: How effective EsViT is when transferring to downstream tasks? **Q3**: What are the design choices and empirical contributions of $\mathcal{L}_R$? **Q4**: When does the intriguing property of self-supervised Transformers exist, including learned correspondence and attentions?

### 4.1 COMPARISONS WITH PRIOR ART ON IMAGENET

We report top-1 linear probe and $k$-NN accuracy on the ImageNet validation set. Table 1 presents comparisons with SoTA SSL systems across various architectures. Please refer to Figure 1 for comparisons over scaling parameter counts and throughput. Our findings are summarized below.

**Comparisons with self-supervised Transformers.**    The DINO- and MoCo-based ViT has higher accuracy and smaller models than iGPT, under the same linear probing protocol and training data.

| Method | #Parameters ↓ | Throughput ↑ | Linear ↑ | $k$-NN ↑ |
|---|---|---|---|---|
| *SoTA SSL methods with Big ConvNets* | | | | |
| SwAV, RN50w5 (Caron et al., 2020) | 586 | 76 | 78.5 | 67.1 |
| BYOL, RN200w2 (Grill et al., 2020) | 250 | 123 | 79.6 | 73.9 |
| SimCLR-v2, RN152w3+SK (Chen et al., 2020d) | 794 | 46 | 79.8 | 73.1 |
| *Skyline methods with excessively long sequences for self-attentions* | | | | |
| DINO, DeiT-S/8 (Caron et al., 2021) | 21 | 180 | 79.7 | 78.3 |
| DINO, ViT-B/8 (Caron et al., 2021) | 85 | 63 | 80.1 | 77.4 |
| MoCo-v3, ViT-B-BN/7 (Chen et al., 2021) | 85 | ∼63 | 79.5 | - |
| MoCo-v3, ViT-L-BN/7 (Chen et al., 2021) | 304 | ∼17 | 81.0 | - |
| iGPT, iGPT-XL (Chen et al., 2020b) | 6801 | - | 72.0 | - |
| EsViT, Swin-T/$W=14$ | 28 | 660 | 78.7 (77.9) | 77.0 (75.5) |
| EsViT, Swin-S/$W=14$ | 49 | 383 | 80.8 (79.4) | 79.1 (77.3) |
| EsViT, Swin-B/$W=14$ | 87 | 254 | **81.3** (80.5) | **79.3** (78.3) |
| *Transformer-based SSL, with moderate sequence length for self-attentions* | | | | |
| Masked Patch Pred., ViT-B/16 (Dosovitskiy et al., 2021) | 85 | 312 | 79.9[†] | - |
| DINO, DeiT-S/16 (Caron et al., 2021) | 21 | 1007 | 77.0 | 74.5 |
| DINO, ViT-B/16 (Caron et al., 2021) | 85 | 312 | 78.2 | 76.1 |
| MoCo-v3, ViT-B/16 (Chen et al., 2021) | 85 | 312 | 76.7 | - |
| MoCo-v3, ViT-H-BN/16 (Chen et al., 2021) | 632 | ∼32 | 79.1 | - |
| MoBY, Swin-T (Xie et al., 2021b) | 28 | 808 | 75.1 | - |
| EsViT, Swin-T | 28 | 808 | 78.1 (77.0) | 75.7 (74.2) |
| EsViT, Swin-S | 49 | 467 | 79.5 (79.2) | 77.7 (76.8) |
| EsViT, Swin-B | 87 | 297 | **80.4** (79.6) | **78.9** (77.7) |

Table 1: Comparison with SoTA across different architectures on ImageNet linear probing. EsViT with $\mathcal{L}_L + \mathcal{L}_R$ is reported, while EsViT with only $\mathcal{L}_R$ is shown in parentheses. $W = 14$ is the window size, otherwise the default $W = 7$. ViT-BN is ViT that has BatchNorm (Frankle et al., 2020), and "/$P$" denotes a patch size of $P \times P$. "∼" indicates through-puts estimated by comparing different papers, detailed in Appendix. [†] The mask patch prediction in (Dosovitskiy et al., 2021) is pre-trained on JFT-300M and end-to-end fine-tuned in ImageNet, which we append as a reference.

At the similar level of model size and compute complexity, the proposed EsViT improve SoTA methods DINO/MoCo-v3 by a large margin: EsViT (Swin-B) outperforms DINO (ViT-B/16) by 2.2% linear probe accuracy and 2.8% $k$-NN accuracy in absolute values. EsViT (Swin-B) even performs slightly better than DINO (ViT-B/8) (0.3% higher linear probe accuracy and 1.5% higher $k$-NN accuracy), with $4\times$ higher throughput. MoBY (Xie et al., 2021b) is a con-current work that investigates multi-stage ViT in SSL. With the same architecture Swin-T, our EsViT pre-training tasks significantly outperform MoBY, showing 3% higher accuracy. In EsViT, longer sequences in self-attention is implemented by increasing the window size. We experiment this by considering a window size of $W=14$. Overall, the proposed EsViT (Swin-B/$W=14$) shows the best performance (top-1 accuracy 81.3%, top-5 accuracy 95.5%, $k$-NN accuracy 79.3%), compared with all systems, and is $3.5\times$ parameter-efficient and has at least $10\times$ higher throughput than previous SoTA MoCo-v3.

**Comparisons with big ConvNets.** We compare with the SoTA big ResNets reported by SimCLR-v2 (Chen et al., 2020d), BYOL (Grill et al., 2020) and SwAV (Caron et al., 2020). Among them, the best accuracy 79.8% under the linear probing protocol is reported by SimCLR-v2 with SK-ResNet, where Selective Kernel (SK) (Li et al., 2019c) is a form of attention to enhance CNNs. It is clear in Figure 1 (b) that all ConvNets-based SSL methods show an envelope in the regime of scaling up model sizes after passing 500M. EsViT achieves better accuracy than their highest envelope, with $16\times$ less model parameters and $8\times$ higher throughput.

### 4.2 TRANSFER LEARNING

We also conduct transfer learning in downstream tasks to evaluate the quality of learned representations. Two sets of tasks are considered:

- *Classification on a suite of 18 small datasets*. As exemplified in (Radford et al., 2021), it is a common and clean approach to evaluate a learned representation by fitting a linear classifier on the representation and measuring its performance across multiple datasets. We study 18 datasets used in (Radford et al., 2021). Automatic hyper-parameter tuning is considered to ensure fairness of comparison. Besides averaged `scores`, we report `# wins` as the number of datasets on which the model outperforms its supervised counterpart. Detailed dataset description and settings are in Appendix.

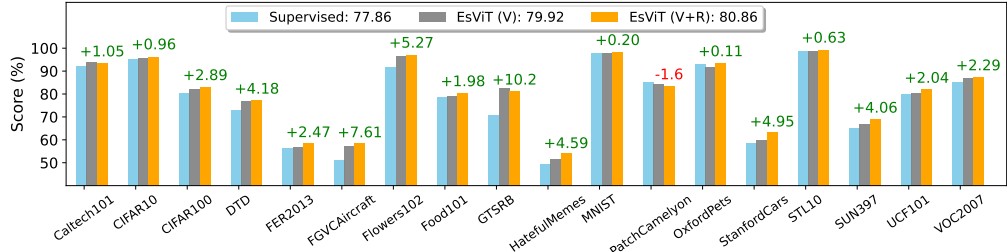

Figure 3: Linear probing on 18 downstream datasets. Averaged scores are reported for each method. EsViT outperforms its supervised counterpart on 17 out of 18 datasets.

| | AP$^{bb}$ | AP$^{bb}_{50}$ | AP$^{bb}_{75}$ |
|---|---|---|---|
| Sup. | 46.0 | 68.1 | 50.3 |
| EsViT | 46.2 (46.2) | 68.0 (67.9) | 50.6 (50.5) |
| | AP$^{mb}$ | AP$^{mb}_{50}$ | AP$^{mb}_{75}$ |
| Sup. | 41.6 | 65.1 | 44.9 |
| EsViT | 41.6 (41.7) | 64.9 (64.8) | 44.8 (45.1) |

Table 2: COCO Detection & Segmentation.

| Pre-train Data | ImageNet-1K | | 18 Datasets | |
|---|---|---|---|---|
| | Linear | $k$-NN | Scores | # Wins |
| Supervised | - | - | 77.29 | - |
| ImageNet-1K | 78.0 (77.1) | 75.7 (73.7) | 80.66 | 16 |
| WebVision-v1 | 75.9 (75.4) | 71.2 (69.4) | 80.00 | 14 |
| OpenImages-v4 | 70.6 (69.6) | 62.0 (60.3) | 77.97 | 10 |
| ImageNet-22K | 75.0 (73.5) | 67.9 (66.1) | **81.03** | **17** |

Table 3: Impact of the pre-train datasets.

- *Detection and segmentation on COCO.* Different from previous monolithic self-supervised ViT, the multi-stage architecture in EsViT can be readily used for dense visual tasks that require hierarchical feature representations.

**Comparison with supervised counterparts.** We compare with the supervised-learning Swin, whose checkpoints are downloaded from the official codebase[2]. Figure 3 shows the classification results of Swin-S, EsViT consistently outperforms its supervised variant, often by a large margin. Similar conclusions are drawn for other model sizes. On COCO detection and segmentation task, however, EsViT shows comparable results with the variant with $\mathcal{L}_V$ only (shown in parentheses) and the supervised counterpart (Swin-T trained with $3\times$ schedule), as shown in Table 2. We hypothsize this is related to the non-constrastive nature of EsViT, as explained later.

**Effects of larger, less-curated pre-train datasets.** The performance of Transformer-based SSL research has thus far been limited to highly curated pre-train data such as ImageNet-1K. To push the frontier in leveraging large amounts of unlabeled data, we explore the effects of pre-training from larger, less-curated image datasets: WebVision-v1 (Li et al., 2017), OpenImages-v4 (Kuznetsova et al., 2020) and ImageNet-22K (Deng et al., 2009), described in Appendix. The pre-train epochs on different datasets are adjusted so that all models see a similar number of augmented views. We summarize the results in Table 3 and would like to emphasize the following findings. First, $\mathcal{L}_R$ improves $\mathcal{L}_V$ (shown in parentheses) on all datasets. Second, all EsViT pre-trained checkpoints outperform supervised checkpoint in downstream classification tasks, but performance varies a lot, with ImageNet-22K checkpoint showing the best transfer ability. Third, ImageNet-1K pre-trained model shows the best ImageNet-1K linear probe performance. We hypothesize that it is not only the size of pre-train dataset matters, but also the distribution of image classes matters: more diverse and well-balanced distribution results in a stronger generalization ability.

## 4.3 DISCUSSION ON THE NON-CONTRASTIVE REGION-MATCHING TASK

**Compatibility with various network architectures.** We investigate ResNet-50 and different efficient sparse Transformers in Table 4. DeiT is shown as a baseline reference. Batch size = 1024 in this experiment. To ensure fair comparison, we modify all into a 4-stage architecture with the number of Transformer layers in each stage as 2-2-6-2. We see that $\mathcal{L}_R$ improves all network architectures, including ResNet-50, Swin (Liu et al., 2021), ViL (Zhang et al., 2021), CvT (Wu et al., 2021) and PvT (Wang et al., 2021). Though directly adding $\mathcal{L}_R$ to monolithic ViT is computationally infeasible, we uniformly sampled top-layer grid features of DeiT and then add $\mathcal{L}_R$, but did not observe performance improvement. This is partly because the monolithic ViT itself already has a good corresponding ability, an extra region-matching task does not provide new learning signals. As

---

[2]https://github.com/microsoft/Swin-Transformer

| Method | #Param. | Im./s | Pre-train tasks | Linear | $k$-NN |
|---|---|---|---|---|---|
| DeiT | 21 | 1007 | $\mathcal{L}_V$ | 75.9 | 73.2 |
| R-50 | 24 | 1237 | $\mathcal{L}_V$ | 75.3[†] | 67.5[†] |
| | | | $\mathcal{L}_V$ | 75.0 | 69.3 |
| | | | $\mathcal{L}_V+\mathcal{L}_R$ | **75.7** | **71.2** |
| Swin | 28 | 808 | $\mathcal{L}_V$ | 77.1 | 73.7 |
| | | | $\mathcal{L}_V+\mathcal{L}_R$ | **77.6** | **75.4** |
| ViL | 28 | 386 | $\mathcal{L}_V$ | 77.3 | 73.9 |
| | | | $\mathcal{L}_V+\mathcal{L}_R$ | **77.5** | **74.5** |
| CvT | 29 | 848 | $\mathcal{L}_V$ | 77.6 | 74.8 |
| | | | $\mathcal{L}_V+\mathcal{L}_R$ | **78.5** | **76.7** |
| PvT | 24 | 851 | $\mathcal{L}_V$ | 75.4 | 72.0 |
| | | | $\mathcal{L}_V+\mathcal{L}_R$ | **76.3** | **72.9** |

Table 4: Different architectures with and without $\mathcal{L}_R$. DeiT and ResNet-50 are shown as references. [†] Numbers reported in (Caron et al., 2021).

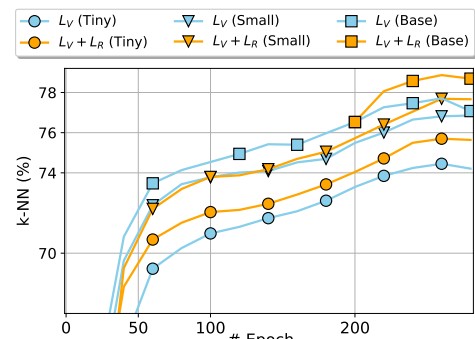

Figure 4: Learning curves of different pre-training tasks. For Base model, $\mathcal{L}_R$ is added from the 200th epoch.

| | | | | ImageNet-1K | | COCO | |
|---|---|---|---|---|---|---|---|
| Types | Methods | #Epochs | #Views | Linear | $k$-NN | AP$^{bb}$ | AP$^{mb}$ |
| | Supervised | | | - | - | 38.2 | 33.3 |
| Contrastive | MoCo-v2 | 200 | 2 | 67.5 | 55.6 | 38.7 | 33.9 |
| | DenseCL | 200 | 2 | 63.6 (-3.9) | 48.6 (-7.0) | 39.1 (+0.4) | 34.2 (+0.3) |
| Non-Contrastive | $\mathcal{L}_V$ | 200 | 2 | 69.2 | 59.9 | 37.8 | 33.1 |
| | $\mathcal{L}_V+\mathcal{L}_R$ | 200 | 2 | 69.9 (+0.7) | 61.7 (+1.8) | 38.0 (+0.2) | 33.2 (+0.1) |

*Pre-training **ResNet50** in different settings*

Table 5: Comparison between contrastive and non-contrastive region-matching tasks.

compared in Appendix Table 12 with the ResNet-50 backbone, EsViT learning method shows the highest accuracy, compared with existing SSL methods.

**Model scaling with $\mathcal{L}_R$.** We compare the pre-training objective with and without $\mathcal{L}_R$ in Table 1. Across different model scales and window sizes, the proposed region level $\mathcal{L}_R$ can consistently improve the performance. The gains can be clearly seen by $k$-NN accuracy (around 1-2%), where no additional tuning is needed as in linear probe. Figure 4 demonstrates that $\mathcal{L}_R$ helps model convergence, and can be used as a drop-in to improve models trained with the view level task.

**Contrastive *vs* Non-contrastive region-matching tasks.** The proposed $\mathcal{L}_R$ adds a non-contrastive region-matching task to the non-contrastive view-level task $\mathcal{L}_V$; On the contrary, DenseCL adds a contrastive region-matching task to the contrastive view-level task MoCo-v2. In Table 5, we compare four methods in the same setting with ResNet-50. DenseCL improves dense visual prediction performance, but hurts classification performance. $\mathcal{L}_R$ improves both tasks, especially the classification performance. One limitation is that the non-contrastive methods show lower performance in dense prediction tasks, this is consistent with the observations for BYOL in (Wang et al., 2020b). The simple $\mathcal{L}_R$ shows the best ImageNet accuracy compared with all sophisticated region-level tasks in this 200-epoch setting in Appendix Table 7, and the best overall accuracy in Table 12. It indicates that $\mathcal{L}_R$ well serves our goal in building efficient SoTA SSL systems.

**Design choices of $\mathcal{L}_R$.** We ablate a couple of choices in constructing $\mathcal{L}_R$ in Eq. (2). (*i*) `Softmax` *vs* `MSE`. One alternative way to measure the distance between two projected vectors is `MSE`, as employed in the popular non-contrastive SSL algorithm BYOL (Grill et al., 2020). When adding region-matching tasks to BYOL and pre-training 50 epochs, `Softmax` and `MSE` yield $k$-NN accuracy of 37.2% and 34.9%, while the baseline BYOL yields 33.1%. We also replace the region-matching metric in EsViT as `MSE`, yielding $k$-NN accuracy 72.6%, which lower than the view-level task only (74.2%). These results show that `Softmax` is essential in $\mathcal{L}_R$. (*ii*) Optimal Transport (OT) *vs* Simple Argmax. To avoid heavy computational overhead, a simple feature-level argmax solution is considered in Eq. (2) to pair two local regions. To study the impact of high region-matching quality, we consider OT. Empirically, we observe OT yields slightly higher $k$-NN accuracy at the early stage, but the gain is diminished in the end. Considering the extra computational cost of solving OT with an inner loop in sinkhorn algorithm (Cuturi, 2013), we opt for simple argmax in our experiments.

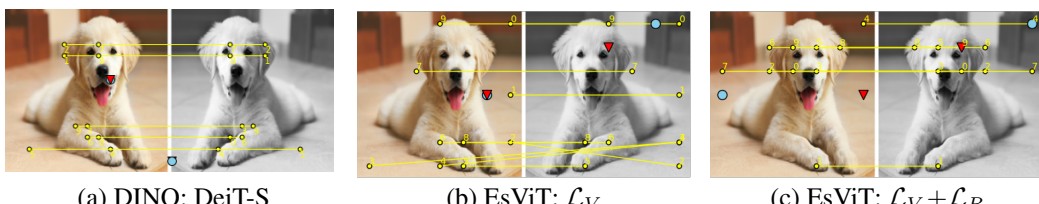

(a) DINO: DeiT-S      (b) EsViT: $\mathcal{L}_V$      (c) EsViT: $\mathcal{L}_V + \mathcal{L}_R$

Figure 5: The learned correspondences. **Yellow** lines are the top-10 correspondences between two views, where the numbers indicates the rankings of similarity scores, yellow dots with the same number are paired.

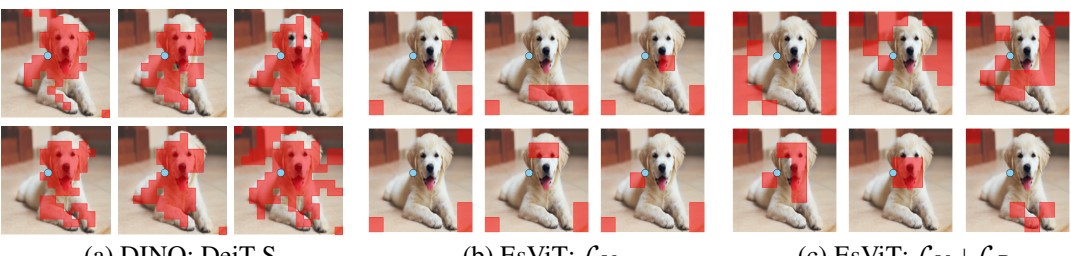

(a) DINO: DeiT-S      (b) EsViT: $\mathcal{L}_V$      (c) EsViT: $\mathcal{L}_V + \mathcal{L}_R$

Figure 6: Visualization of the the learned attention map for different heads in the last layer. The query is the **blue** dot in the center of the images. We visualize masks (as **red**) obtained by thresholding the self-attention maps to keep 60% of the probability mass. Note that all 6 heads are visualized for DINO with DeiT-S, and 6 out of 24 heads in EsViT are chosen to visualize (ranked by entropy values). Please see enlarged pictures with all heads in Appendix.

## 4.4 QUALITATIVE STUDIES

**Visualization of correspondences.** Given two views of the same image, we use the pre-trained backbone to extract the top-layer features $z_1$ and $z_2$. For each feature vector in $z_1$, we find the feature vector in $z_2$ that best matches it in terms of highest cosine similarity, as defined in Equation (2). In Figure 5, we show the top-10 correspondences between two views for three methods. In Figure 5 (b), EsViT with $\mathcal{L}_V$ tends to identify pairs in the background as the most matched ones (and in a wrong way in this example). This could be a valid solution to $\mathcal{L}_V$, as the invariance in the level of aggregated global features does not necessarily induce invariances in the local region level. This is significantly alleviated with $\mathcal{L}_R$ (shown in Figure 5 (c)), a task that implicitly requires local matching.

Surprisingly, DINO is able to learn good correspondences even without the region-level matching task. To the best of our knowledge, this is a previously unreported intriguing property of self-supervised Transformers with monolithic architectures: good semantic correspondences are automatically learned. We hypothesize that features at lower layers (image patch itself in the extreme case) can directly pass to higher layers, and the former regularizes the latter to remain discriminative. Nevertheless, the proposed $\mathcal{L}_R$ can dramatically reduce the issue, and is good remedy to rescue the loss of semantic correspondence for the multi-stage architecture. In Appendix, we quantitatively measures the correspondence learning ability of these SSL methods on ImageNet validation dataset, the observations are consistent: $\mathcal{L}_R$ improves the matching accuracy from 66% to 91%.

**Visualization of attention maps.** We look at the self-attention in the different heads of the last layer in Figure 6. A local region on the edge of the main object is employed as query, and the attended regions are highlighted in red for those the query's top 60% mass are assigned. In Appendix, we visualize more examples with different query positions. DINO tends to automatically learn class-specific attention maps leading to foreground object segmentation, regardless of its query located in foreground or background. This is probably because main objects remain as the major invariance factor in different augmented views. This property is lost when a multi-stage architecture is employed, as shown in EsViT with $\mathcal{L}_V$. These patterns are consistent for different heads. After introducing $\mathcal{L}_R$ for EsViT, we note that the attention maps become more diverse in different heads, *i.e.,* entropy values of attentions get more skewed, and attended regions are more different. This is perhaps because $\mathcal{L}_R$ requires each region to consider many matching tasks to regions in different augmented views, each head automatically learns to distribute the tasks and complete a few of them.

## 5 CONCLUSIONS

In this paper, we first discover the automatic correspondence learning property of self-supervised monolithic Transformers. Inspired by this, we present efficient self-supervised vision Transformers (EsViT) to with two major insights: a multi-stage Transformer architecture with sparse self-attentions, and a non-contrastive region-matching pre-training task. The synergy of both helps EsViT reach the SoTA performance of SSL vision systems with significantly less compute and smaller model size. Our study also reveals that exploration of effective solutions to learn from larger and less curated pre-training data in the wild is a key but less studied factor in paving the way toward the scaling success of SSL vision systems.

## ETHICS STATEMENT

Though self-supervised learning (SSL) has great potentials to learn powerful representation without human annotation, the existing techniques to build SoTA SSL vision systems tend to be **Red AI** (Schwartz et al., 2020): it could be environmentally unfriendly and the computational cost is extensively high. The required training resource is typically not accessible for a lab environment (thus raising barriers to participation in AI research). For example, the prior art MoCo-v3 has greatly pushes the performance limit of SSL system (Chen et al., 2021). The authors kindly reported that "it (MoCo-v3, ViT-H) takes 9.8 hours per 100 epochs using 512 TPUs. This is a gigantic scale of training: for the 300-epoch ViT-H, this amounts to $\sim$625 TPU days, or $\sim$1.7 TPU years of training." The SoTA model MoCo-v3 with ViT-BN-L/7 should have a higher cost than this. Even for a smaller model ViT-B, "it takes 24 hours in 128 GPUs (vs. 2.1 hours in 256 TPUs)". Hence, improving the efficiency of building SoTA SSL systems is of high value for the community and society to achieve **Green AI** (Schwartz et al., 2020).

To this end, we propose EsViT to provide more affordable and efficient solutions for the community to experiment and explore the directions of SoTA SSL in computer vision. Our EsViT model shows the best ImageNet linear probe performance compared with all existing SSL vision systems, and is $3.5\times$ parameter-efficient and has $10\times$ higher throughput than previous SoTA. This efficiency gain can significantly decrease its carbon footprint and increase its inclusivity, encouraging more researchers to participate the study of the SSL topic.

## REPRODUCIBILITY STATEMENT

Our paper provides comprehensive empirical studies on the EsViT algorithm. We provide PyTorch-style pseudo-code in Appendix. We also include an example code with instruction as supplementary material to ensure the reproducibility. For empirical results on both various network architecture and large-scale datasets, we provide detailed hyper-parameter specifications. We will release the pre-trained checkpoints and codebase for the research community for reproducible research.

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

# A  METHODS

## A.1  ALGORITHMS

We summarize the training algorithm procedure of EsViT with $\mathcal{L}_V + \mathcal{L}_R$ in Algorithm 1. To clearly outline the main idea of the algorithm, we show the algorithm for two augmented views. For the full algorithm to deal with multi-crop, please refer to our codebase. In Algorithm 1, for a mini-batch of size $n$, the teacher/student network consists of three output variables: (1) $p \in \mathbb{R}^{n \times K}$ is the probability vector for the view-level representation, output by an MLP head. (2) $z \in \mathbb{R}^{n \times T \times P}$ is the feature map, containing $T$ region-level features of dimension $P$. (3) $pz \in \mathbb{R}^{n \times T \times K}$ are probability vectors of $z$, output by a different MLP head.

---

**Algorithm 1:** EsViT with $\mathcal{L}_V + \mathcal{L}_R$, pseudocode with 2-crop.

```
    # gs, gt:   student and teacher networks
    # Cv, Cr:   view and region center (K)
    # tmp_s, tmp_t:  student and teacher temperatures
    # a, b:  network and center momentum rates.
    # n:  batch size, K: MLP-head-projected probability vector length,
      T: last layer feature map length, P: last layer feature vector
      length
 1  gt.params = gs.params
    # The main training loop
 2  for x in loader:
 3      x1, x2 = augment(x), augment(x)   # two random views
 4
        # student output, p:n×K, pz:n×T×K, z:n×T×P
 5      p_s1, pz_s1, z_s1 = gs(x1)
 6      p_s2, pz_s2, z_s2 = gs(x2)
        # teacher output, p:n×K, pz:n×T×K, z:n×T×P
 7      p_t1, pz_t1, z_t1 = gt(x1)
 8      p_t2, pz_t2, z_t2 = gt(x2)
 9
        # view-level loss
10      loss_v = Hv(p_s1, p_t2)/2 + Hv(p_s2, p_t1)/2
        # region-level loss
11      loss_r = Hr(pz_s1, pz_t2, z_s1, z_t2)/2 + Hr(pz_s2, pz_t1, z_s2, z_t1)/2
12      loss = loss_v/2 + loss_r/2
13      loss.backward()  # back-propagate
14
        # update student, teacher and centers
15      update(gs)                                  # AdamW for student
16      gt.params = a * gt.params + (1 - a) * gs.params   # EMA for teacher
17      Cv = b * Cv + (1 - b) * cat([p_t1, p_t2].mean(0))    # EMA for view center
18      Cr = b * Cr + (1 - b) * cat([pz_t1, pz_t2].mean(0))  # EMA for region center
19
    # The view-level loss function
20  def Hv(s, t):
21      t = t.detach()  # stop gradient
22      s = softmax(s / tmp_s, dim=-1)
23      t = softmax((t - Cv) / tmp_t, dim=-1)
24      return - (t * log(s)).sum(dim=-1).mean()
    # The region-level loss function
25  def Hr(ps, pt, zs, zt):
26      pt = pt.detach()                                # stop gradient
27      ps = softmax(ps / tmp_s, dim=-1)                # n×T×K
28      pt = softmax((pt - Cr) / tmp_t, dim=-1)         # n×T×K
29      sim_matrix = torch.matmul(zs , zt.permute(0, 2, 1))  # n×T×T
30      sim_idx = sim_matrix.max(dim=-1)[1].unsqueeze(2)     # n×T×1
31      pt_idxed = torch.gather(pt, 1, sim_idx.expand(-1, -1, pt.size(2)))
32      return - (pt_idxed * log(ps)).sum(dim=-1).mean()
```

---

## A.2 NETWORK ARCHITECTURE CONFIGURATIONS AND IMPLEMENTATION DETAILS

Inspired by great successes of the multi-stage ConvNet architecture such as VGG (Simonyan & Zisserman, 2014)/ResNets (He et al., 2016) for computer vision, the multi-stage Transformer-based networks have been explored very recently in the supervised learning setting (Vaswani et al., 2021; Wang et al., 2021; Liu et al., 2021; Zhang et al., 2021; Wu et al., 2021). In multi-stage vision Transformers, since a larger number of patches is often produced at the early stages, an efficient Transformer with sparse self-attentions is considered to reduce the computational complexity. The basic idea is to split the feature maps into non-overlapping local windows (with size $W \times W$), and self-attention is performed within each local window. This however has one drawback that features in different local windows cannot interact. Various methods have been proposed to best approximate full-attention, with different trade-off between accuracy and efficiency.

We briefly describe three schemes as follows, and benchmark them in the experiments. (*i*) *Swin Transformer* (Liu et al., 2021): A shifted window partitioning approach is proposed, which alternates between two partitioning configurations in consecutive Transformer blocks, so that each local feature is grouped into different windows in self-attentions. (*ii*) *Vision Longformer (ViL)* (Zhang et al., 2021): Features in each local window are further allowed to attend all features in the 8-neighboring windows. (*iii*) *Convolution vision Transformer (CvT)* (Wu et al., 2021): Features in neighboring windows are considered in the convolutional projection in self-attentions.

The window size is set to $W = 7$ by default. The query dimension of each head in self-attentions is $d = 32$, and the hidden layer width of each MLP is $4\times$ of its input's width, for all experiments. The architecture configurations of model variants employed in the experiments are summarized in Table 6. Some notable implementation detailed are described as follows:

- The three configurations Swin-T, Swin-S and Swin-B indicate Tiny, Small, and Base models, respectively, which are almost identical to the original implementation (Liu et al., 2021), except that we add special treatments to deal with input augmented views of different resolutions, when the resolution (feature map size more specifically) is not divisible by the window size (*i.e.,* resolution 96 and window size=7 or 14).

- Swin-T and Swin-S with window size $W = 14$ are customized by us to allow full self-attention in stage 3 (where the majority of model capacity is allocated to) and stage 4, to study the impact of longer sequences in EsViT.

- In the original ViL (Zhang et al., 2021) and CvT (Wu et al., 2021) papers, different positional embedding strategies and multi-stage network configurations were employed. We modify them by only utilizing relative position bias and their proposed sparse self-attention mechanisms, and create a similar 4-stage architecture with Swin-T for fair comparison.

**Relative Position Bias.** To facilitate SSL, we consider relative position bias (Liu et al., 2021) to characterize the spatial information between features for the three efficient Transformers aforementioned, and do not use absolute position embeddings. This is because augmented views of varied resolutions can be cropped from anywhere in an image in SSL, maintaining the relative positions is easy in implementation, and is largely sufficient for invariance learning among these views.

## B RELATED WORK

**Self-supervised ConvNets.** ConvNets-based SSL has been extensively studied in the literature. Based on the pre-training tasks, they can be broadly categorized into three classes: Handcrafted pretext tasks (Doersch et al., 2015; Noroozi & Favaro, 2016; Pathak et al., 2016; Gidaris et al., 2018; Zhang et al., 2016; Larsson et al., 2016; Zhang et al., 2017; Pu et al., 2016; Donahue & Simonyan, 2019), contrastive learning (Dosovitskiy et al., 2015; Zhuang et al., 2019; Oord et al., 2018; Hjelm et al., 2018; Bachman et al., 2019; He et al., 2020; Chen et al., 2020c; Grill et al., 2020) and prototype learning (Caron et al., 2018; 2020; Li et al., 2020b; Xie et al., 2016; Yang et al., 2016; Ji et al., 2019; Zhan et al., 2020). It is also known that data augmentations play a crucial role in SSL pipeline (Chen et al., 2020e; Caron et al., 2020; Tian et al., 2020; Li et al., 2020a). The impact of pre-training dataset size/quality is explored for ConvNets in SSL (Goyal et al., 2021; Yonglong et al., 2021). To date, the search of best pre-taining tasks/datasets and augmentations are based on CNNs. Among them,

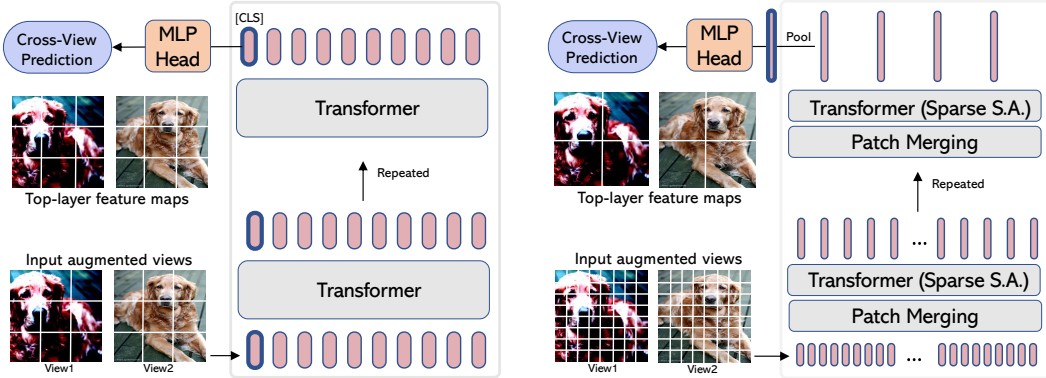

(a) Baseline monolithic architecture      (b) Proposed multi-stage architecture

Figure 7: Architecture comparison. (a) The monolithic transformer. For all layers, the transformer blocks share the same network configurations and input token sequence sizes are the same. (b) The multi-stage Transformer organizes an input image into a long sequence of smaller patches, sparse self-attentions (S.A.) are utilized at early stages to maintain model expressiveness while reducing computational complexity; The neighboring tokens at an intermediate layer are gradually merged, constituting a short sequence to ease the compute burden of self-attention at late stages.

|  | Stage 1 | Stage 2 | Stage 3 | Stage 4 |
|---|---|---|---|---|
| Merging Rate | 4× | 8× | 16× | 32× |
| Feature Map | 56 × 56 | 28 × 28 | 14 × 14 | 7 × 7 |
| Swin-T, $W=7$ | concat 4 × 4 , 96-d, LN
[ window size : 7 × 7
96-d (3 heads) ] ×2 | concat 2 × 2, 192-d, LN
[ window size : 7 × 7
192-d (6 heads) ] ×2 | concat 2 × 2, 384-d, LN
[ window size : 7 × 7
384-d (12 heads) ] ×6 | concat 2 × 2, 768-d, LN
[ window size : 7 × 7
768-d (24 heads) ] ×2 |
| Swin-S, $W=7$ | concat 4 × 4 , 96-d, LN
[ window size : 7 × 7
96-d (3 heads) ] ×2 | concat 2 × 2, 192-d, LN
[ window size : 7 × 7
192-d (6 heads) ] ×2 | concat 2 × 2, 384-d, LN
[ window size : 7 × 7
384-d (12 heads) ] ×18 | concat 2 × 2, 768-d, LN
[ window size : 7 × 7
768-d (24 heads) ] ×2 |
| Swin-B, $W=7$ | concat 4 × 4 , 128-d, LN
[ window size : 7 × 7
128-d (4 heads) ] ×2 | concat 2 × 2, 256-d, LN
[ window size : 7 × 7
256-d (8 heads) ] ×2 | concat 2 × 2, 512-d, LN
[ window size : 7 × 7
512-d (16 heads) ] ×18 | concat 2 × 2, 1024-d, LN
[ window size : 7 × 7
1024-d (32 heads) ] ×2 |
| Swin-T, $W=14$ | concat 4 × 4 , 96-d, LN
[ window size : 14 × 14
96-d (3 heads) ] ×2 | concat 2 × 2, 192-d, LN
[ window size : 14 × 14
192-d (6 heads) ] ×2 | concat 2 × 2, 384-d, LN
[ window size : 14 × 14
384-d (12 heads) ] ×6 | concat 2 × 2, 768-d, LN
[ window size : 7 × 7
768-d (24 heads) ] ×2 |
| Swin-S, $W=14$ | concat 4 × 4 , 96-d, LN
[ window size : 14 × 14
96-d (3 heads) ] ×2 | concat 2 × 2, 192-d, LN
[ window size : 14 × 14
192-d (6 heads) ] ×2 | concat 2 × 2, 384-d, LN
[ window size : 14 × 14
384-d (12 heads) ] ×18 | concat 2 × 2, 768-d, LN
[ window size : 7 × 7
768-d (24 heads) ] ×2 |
| ViL-T, $W=7$ | concat 4 × 4 , 96-d, LN
[ window size : 7 × 7
96-d (3 heads) ] ×2 | concat 2 × 2, 192-d, LN
[ window size : 7 × 7
192-d (3 heads) ] ×2 | concat 2 × 2, 384-d, LN
[ window size : 7 × 7
384-d (6 heads) ] ×6 | concat 2 × 2, 768-d, LN
[ window size : 7 × 7
768-d (12 heads) ] ×2 |
| CvT-T, $W=7$ | concat 4 × 4 , 64-d, LN
[ window size : 7 × 7
64-d (1 head) ] ×2 | concat 2 × 2, 192-d, LN
[ window size : 7 × 7
192-d (3 heads) ] ×2 | concat 2 × 2, 384-d, LN
[ window size : 7 × 7
384-d (6 heads) ] ×6 | concat 2 × 2, 768-d, LN
[ window size : 7 × 7
768-d (12 heads) ] ×2 |

Table 6: Model configurations considered in our experiments.

SimCLR-v2 (Chen et al., 2020d), BYOL (Grill et al., 2020) and SwAV (Caron et al., 2020) achieve the highest ImageNet linear probe performance with large ConvNet architectures. The performance tends to saturate with an increasingly growing model size, raising a question if ConvNets reach a limit in SSL.

**Transformers for vision.** Vision Transformers (ViT) (Dosovitskiy et al., 2021) shows the great potentials of generalizing Transformers for computer vision, by achieving compelling accuracy in supervised learning, especially with large-scale data and high capacity models. DeiT (Touvron et al., 2020) further provides an effective ViT training strategy to ease the adaption of Transformers for practitioners. Transformers have also been applied to other vision tasks, ranging from low-level tasks such as image generation (Parmar et al., 2018; Chen et al., 2020b) and enhancement (Chen et al., 2020a; Yang et al., 2020), to high-level tasks such as object detection (Carion et al., 2020; Zhu et al., 2020; Zheng et al., 2020; Dai et al., 2020) and segmentation (Wang et al., 2020a;c), and to vision-language tasks (Lu et al., 2019; Tan & Bansal, 2019; Chen et al., 2019; Su et al., 2019;

| Name | Framework | Pre-train Task Description | Major Motivation | Major Downstream Task Performance | ImageNet |
|------|-----------|---------------------------|------------------|-----------------------------------|----------|
| PIRL | Contrastive | Jigsaw pretext task in a way that encourages the image representations to be invariant to the image patch perturbation | General-purpose visual backbone learning Mostly for image classification. | Image classification | 63.6% |
| DenseCL | Contrastive | A pairwise contrastive (dis)similarity loss at the patch level between two views. A queue of negative sample features from other images is maintained | Improving dense visual prediction task performance | Improving object detection performance. | 63.6% |
| DetCo | Contrastive | Multi-level features with three contrastive tasks between global images and local patches are considered: global-global, global-local, local-local. | Improving dense visual prediction task performance | Improving object detection performance. DetCo achieves the best performance trade-off on both classification and detection. | 68.6% |
| PixPro | Contrastive | Features from the two views are encouraged to be consistent between a regular patch representation and a smoothed patch representation within the same image. | Improving dense visual prediction task performance | Mostly focusing on the improved performance on detection tasks. | 66.3% |
| InstLoc | Contrastive | image instances are pasted at various locations and scales onto background images. The pretext task is to predict the instance category given the composited images and the foreground bounding boxes | Improving dense visual prediction task performance | Mostly focusing on the improved performance on detection tasks. | 61.7% |
| EsViT (ours) | Non-contrastive | A pairwise cross-entropy loss at the patch level between two positive views. No need/interaction with other images in the batch (eg, no negative samples) | Recovering the automatic correspondence learning property of self-supervised monolithic transformers, and thus improving learning efficiency. | Consistently improving image classification tasks. It creates new SoTA 81.3% on ImageNet linear probe accuracy, showing 3.5x parameter-efficient and has 10x higher throughput than previous SoTA MoCo-v3. Reporting 75.7% ImageNet linear probe performance for ResNet-50. | **69.9%** |

Table 7: Discussion of related works on various region-level tasks. The last columns reports the ImageNet linear probe performance for ResNet-50 trained with 2 augmented views for 200 epochs.

Li et al., 2019b;a; Zhou et al., 2020; Li et al., 2020c). Marrying Transformers with multi-stage architectures (Vaswani et al., 2021; Wang et al., 2021; Liu et al., 2021; Zhang et al., 2021; Wu et al., 2021) show higher classification accuracy in supervised learning, and enables applicability of Transformers for a broader range of vision tasks. Given these properties, we believe multi-stage ViT is a must-study baseline for SSL in computer vision.

**Discussion with other region-level tasks.** In Table 7, we compare $\mathcal{L}_R$ against the existing region-level tasks, including PIRL (Misra & Maaten, 2020), DenseCL (Wang et al., 2020b), DetCo (Xie et al., 2021a), InstLoc (Yang et al., 2021), PixPro (Xie et al., 2021c). Most of these region-level tasks improve object detection tasks, but hurt the ImageNet classification accuracy. DetCo achieves the best trade-off: improving the performance of both tasks, but with a sophisticated multi-level, global-local interaction algorithm. With the same number of pre-training epochs and augmented views, EsViT achieve the best ImageNet linear probe accuracy among all region-level tasks, with as minimum computational overhead as possible. This well serves our goal of building efficient SSL SoTA image classification system.

| Data source | Table 2 in DINO (Caron et al., 2021) | Table 3 in MLP-Mixer (Tolstikhin et al., 2021) | Table 1 in Swin (Liu et al., 2021) | Our runs |
|---|---|---|---|---|
| DeiT-S / $P=16$ | 1007 | | 940.4 | |
| DeiT-B / $P=16$ | 312 | | 292.3 | |
| DeiT-S / $P=8$ | 180 | | | |
| DeiT-B / $P=8$ | 63 | | | |
| ViT-B / $P=16$ | 312 | 861 | | |
| ViT-S / $P=16$ | 102 | 280 | | |
| ViT-H / $P=14$ | 32 | 87 | | |
| ViT-L / $P=7$ | 17 | 47[†] | | |
| Swin-T / $W=7$ | 808 | | 755.2 | 726.13 |
| Swin-S / $W=7$ | 467 | | 436.9 | |
| Swin-B / $W=7$ | 297 | | 278.1 | |
| Swin-T / $W=14$ | 660 | | | 593.24 |
| Swin-S / $W=14$ | 383 | | | 344.20 |
| Swin-B / $W=14$ | 254 | | | 228.36 |
| ViL-T / $W=7$ | 386 | | | 346.72 |
| CvT-T / $W=7$ | 848 | | | 761.89 |

Table 8: Throughput estimate and standardization. All numbers in orange are estimated/converted, while numbers in blue are collected from the papers, and numbers in green are runs on our machines. All papers report the throughput of ViT-B or DeiT-B, which are essentially the same model. We use this fact to align the throughput reported in different papers. [†] This number is estimated via the statement in (Chen et al., 2021) that "reducing the patch size to $7 \times 7$ keeps the model size unchanged, but increases FLOPs to $\sim 6\times$". All numbers are standardized into throughput reported by (Caron et al., 2021).

## C    EXPERIMENTS

### C.1    THROUGHPUT ESTIMATE AND CONVERSION

Since different papers report throughput on different hardwares, it is not ready to compare the numbers directly. Noting that all papers report the throughput for ViT-B/DeiT-B, we use this number to align and convert the throughput. In Table 8, we describe our process and results of standardizing the throughput.

### C.2    THE COMPUTATION AND MEMORY OVERHEAD OF THE PROPOSED $\mathcal{L}_R$

We emphasize that adding $\mathcal{L}_R$ to the multi-stage transformer architectures yields acceptable extra computational cost, while adding $\mathcal{L}_R$ directly to the monolithic transformer architectures has a huge computational overhead. To demonstrate this, we report the cost comparisons in Table 9. For each setting, we report [Memory Usage (MB) / Running time per iteration (second/iteration)]. In Table Table 9 (a), when the batch size is gradually increased (*e.g.,* , batch-size=12), the memory cost increases nearly 4 times for monolithic architectures, while increases 1.6 times for multi-stage architectures. Similar trends are shown for training cost per iteration increase ratio (1.47 vs 1.15). This indicates $\mathcal{L}_R$ can more naturally fit multi-stage architectures.

Similarly, we compare computational cost comparisons [Memory Usage (MB) / Running time per iteration (second/iteration)] in Table 9 (b), for other network architecture configurations . From the increased cost ratio, we see that $\mathcal{L}_R$ adds acceptable cost in terms of both memory and training time, compared with the baseline.

### C.3    EXPERIMENTAL SETTINGS OF PRE-TRAINING AND EVALUATION ON IMAGENET

We study unsupervised pre-training performed in ImageNet-1K dataset (Deng et al., 2009) without labels. The default training details are described as follows, mostly following (Caron et al., 2021). We train with the Adamw optimizer (Loshchilov & Hutter, 2018), a batch size of $512$, and total epochs 300. Linear warmup of the learning rate is used during the first 10 epochs, with its base value

|  |  |  | Batch-Size=1 | Batch-Size=8 | Batch-Size=12 |
|---|---|---|---|---|---|
| Monolithic | ViT-S | $\mathcal{L}_V$ | 1391 / 0.209036 | 2735 / 0.234993 | 3533 / 0.238160 |
|  | ViT-S | $\mathcal{L}_V + \mathcal{L}_R$ | 2641 / 0.233150 | 10358 / 0.321155 | 14128 / 0.352339 |
|  | **Increased Cost Ratio** |  | **1.8986 / 1.1153** | **3.7872 / 1.3666** | **3.9988 / 1.4794** |
| Multi-stage | Swin-T | $\mathcal{L}_V$ | 1704 / 0.285451 | 4229 / 0.323884 | 5611 / 0.366367 |
|  | Swin-T | $\mathcal{L}_V + \mathcal{L}_R$ | 2346 / 0.301672 | 6232 / 0.374634 | 8917 / 0.421135 |
|  | **Increased Cost Ratio** |  | **1.3767 / 1.0568** | **1.4736 / 1.2323** | **1.5889 / 1.1494** |

(a) Comparisons for increased batch sizes.

| EsViT | $\mathcal{L}_V$ | $\mathcal{L}_V + \mathcal{L}_R$ | **Increased Cost Ratio** |
|---|---|---|---|
| Tiny (W=7) | 1704 / 0.285451 | 2346 / 0.301672 | **1.3767 / 1.0568** |
| Small (W=7) | 2685 / 0.501876 | 3132 / 0.535203 | **1.1664 / 1.0664** |
| Base (W=7) | 3726 / 0.516058 | 4374 / 0.550617 | **1.1739 / 1.0669** |
| Tiny (W=14) | 2159 / 0.288118 | 2801 / 0.310108 | **1.0890 / 1.0763** |
| Small (W=14) | 3518 / 0.496823 | 4153 / 0.521739 | **1.1805 / 1.0501** |
| Base (W=14) | 5032 / 0.511701 | 5681 / 0.537826 | **1.1289 / 1.0510** |

(b) Comparisons for various network architecture configurations.

Table 9: Computational cost comparisons in the format of [*Memory Usage (MB) / Running time per iteration (second/iteration)*].

| Dataset | Classes | Train size | Test size | Evaluation metric | Source link |
|---|---|---|---|---|---|
| Food-101 | 102 | 75,750 | 25,250 | Accuracy | Tensorflow |
| CIFAR-10 | 10 | 50,000 | 10,000 | Accuracy | TensorFlow |
| CIFAR-100 | 100 | 50,000 | 10,000 | Accuracy | TensorFlow |
| SUN397 | 397 | 19,850 | 19,850 | Accuracy | Tensorflow |
| Stanford Cars | 196 | 8,144 | 8,041 | Accuracy | Stanfold Cars |
| FGVC Aircraft (variants) | 100 | 6,667 | 3,333 | Mean-per-class | FGVC website |
| VOC2007 classification | 20 | 5,011 | 4,952 | 11-point mAP | voc2007 |
| Describable Textures | 47 | 3,760 | 1,880 | Accuracy | TensorFlow |
| Oxford-IIIT Pets | 37 | 3,680 | 3,669 | Mean-per-class | Oxford-IIIT Pet |
| Caltech-101 | 102 | 3,060 | 6084 | Mean-per-class | TensorFlow |
| Oxford Flowers 102 | 102 | 2,040 | 6,149 | Mean-per-class | TensorFlow |
| MNIST | 10 | 60,000 | 10,000 | Accuracy | TensorFlow |
| Facial Emotion Recog. 2013 * | 8 | 32,298 | 3,589 | Accuracy | Kaggle fer2013 |
| STL10 | 10 | 5,000 | 8,000 | Accuracy | TensorFlow |
| GTSRB * | 43 | 26,728 | 12,630 | Accuracy | GTSRB website |
| PatchCamelyon | 2 | 294,912 | 32,768 | Accuracy | TensorFlow |
| UCF101 * | 101 | 9,537 | 3783 | Accuracy | TensorFlow |
| Hateful Memes | 2 | 8,500 | 500 | ROC-AUC | FaceBook |

Table 10: A suite of 18 datasets used in linear probe.* indicates dataset whose train/test size we obtained is slightly different from Table 9 in (Radford et al., 2021).

determined with the linear scaling rule (Goyal et al., 2017): $lr = 0.0005 * \text{batchsize}/256$. After this warmup, the learning rate is decayed with a cosine schedule. We build our systems based on Swin Transformers (Liu et al., 2021) in our experiments. Swin-B has a model size and computation complexity similar to ViT-B/DeiT-B (patch size 16). We also considered Swin-T and Swin-S, which have the complexity that are similar to those of ResNet-50 (DeiT-S) and ResNet-101, respectively. The default window size is $W = 7$.

One major common protocol to evaluate SSL is linear probe on ImageNet-1K, where features are extracted from a frozen backbone, and a supervised linear classifier is trained. For all Transformer models, we use the concatenation of view-level features $\bar{z}$ in the last 4 layers (the results are similar to the use of last 3 or 5 layers in our initial experiments).

| Methods | CLIP ResNet-50 | Supervised ResNet-50 | Supervised[‡] ResNet-50 | Supervised Swin-T | EsViT Swin-T |
|---|---|---|---|---|---|
| Food-101 | 86.4 | 71.3 | 71.3 | 77.4 | 80.0 |
| CIFAR-10 | 88.7 | 91.8 | 91.8 | 94.0 | 95.3 |
| CIFAR-100 | 70.3 | 74.5 | 74.5 | 77.5 | 82.2 |
| SUN397 | 73.3 | 60.5 | 60.3 | 64.3 | 67.6 |
| Stanford Cars | 78.3 | 49.9 | 50.1 | 55.3 | 66.4 |
| FGVC Aircraft (variants) | 49.1 | 48.5 | 48.4 | 51.5 | 61.1 |
| VOC2007 classification | 87.1 | 83.8 | 83.6 | 84.2 | 85.5 |
| Describable Textures | 76.4 | 72.3 | 72.6 | 73.1 | 78.1 |
| Oxford-IIIT Pets | 88.2 | 92.4 | 92.1 | 93.3 | 92.8 |
| Caltech-101 | 89.6 | 90.8 | 90.4 | 90.8 | 93.0 |
| Oxford Flowers 102 | 96.1 | 90.8 | 91.1 | 91.5 | 97.4 |
| MNIST | 98.3 | 98.3 | 98.3 | 98.3 | 98.3 |
| Facial Emotion Recog. 2013 | 64.2 | 54.9 | 55.9 | 55.1 | 59.3 |
| STL10 | 97.2 | 96.4 | 97.0 | 97.9 | 98.9 |
| GTSRB | 82.4 | 70.6 | 75.7 | 72.9 | 84.3 |
| PatchCamelyon | 82.7 | 82.5 | 82.6 | 84.0 | 84.6 |
| UCF101 | 81.6 | 71.2 | 72.1 | 79.0 | 81.1 |
| Hateful Memes | 65.7 | 56.5 | 49.9 | 51.2 | 52.0 |
| Average | 80.86 | 75.39 | 75.43 | 77.29 | 80.99 |

Table 11: The linear probe results on 18 datasets at the scale of ResNet-50/Swin-T. [‡] indicates the results reproduced by us, which verifies that our implementation pipeline is consistent with (Radford et al., 2021).

| Method | View-level | Region-level | Top-1 Accuracy (%) |
|---|---|---|---|
| *Performance comparison of ResNet-50 with 200 epochs and 2 augmented views* | | | |
| MoCo-v2 | Contrastive | - | 67.5 |
| DenseCL | Contrastive | Contrastive | 63.6 |
| DetCo | Contrastive | Contrastive | 68.6 |
| DINO | Non-Contrastive | - | 69.2 |
| EsViT | Non-Contrastive | Non-Contrastive | **69.9** |
| *SoTA performance comparison of ResNet-50 with numbers and settings reported in each paper* | | | |
| MoCo-v2 (800 epochs) | Contrastive | - | 72.2 |
| SwAV (800 epochs, w/ multi-crop) | Contrastive | - | 75.3 |
| Barlow Twins (1000 epochs) | - | - | 73.2 |
| VICReg (1000 epochs) | - | - | 73.2 |
| SimSiam (800 epochs, 2 views) | Non-Contrastive | - | 71.3 |
| BYOL (1000 epochs, w/ multi-crop) | Non-Contrastive | - | 74.3 |
| DINO (300 epochs, w/ multi-crop) | Non-Contrastive | - | 75.0 |
| EsViT (300 epochs, w/ multi-crop) | Non-Contrastive | Non-Contrastive | **75.7** |

Table 12: Linear probe performance of a ResNet-50 network with different SSL methods.

## C.4 COMPARISON WITH A RESNET-50 BACKBONE

To compare our EsViT learning method with other SSL algorithms, we conduct experiments with a ResNet-50 backbone, and show the results in Table 12.

## C.5 LINEAR PROBE ON A SUITE OF SMALL DATASETS

**Datasets.** Table 10 shows details and source of all datasets used for linear probe, including the number of classes, the size of training set and testing set, metrics used in evaluation, as well as a public source of the dataset. Note that original UCF101 dataset is a video dataset. Here the middle frame of each video is extracted to form a classification dataset. There are 3 train/val splits in Tensorflow, we use the first one.

**Automatic hyper-parameter tuning.** We rigorously follow (Radford et al., 2021) to conduct training and evaluation for linear probe on the downstream datasets. We train a logistic regression classifier using scikit-learn's L-BFGS implementation, with maximum $1,000$ iterations, and report the corresponding metric for each dataset. We determine the $L_2$ regularization strength $\lambda$ using a hyperparameter sweep on the validation sets over the range between $10^{-6}$ and $10^6$ , with 96 logarithmically spaced steps. To save compute required for the sweeps, we perform a parametric binary search that starts with $\lambda = [10^{-6}, 10^{-4}, 10^{-2}, 1, 10^2, 10^4, 10^6]$ and iteratively halves the interval around the peak until it reaches a resolution of 8 steps per decade. The hyperparameter sweeps are performed on a validation split of each dataset. For the datasets that contain a validation split in addition to a test split, we use the provided validation set to perform the hyperparameter search, and for the datasets that do not provide a validation split or have not published labels for the test data, we split the training dataset to perform the hyperparameter search. For the final result, we combine the validation split back with the training split and report the performance on the unused split.

**Detailed results.** Only the last layer feature is considered for all models for simplicity, though adding features from more layers may potentially improve the results. Table 11 shows the results for architectures at a similar scale of ResNet-50 or Swin-T. The first two columns are numbers from (Radford et al., 2021). CLIP with ResNet-50 is pre-trained on 400 million image-text pairs. Supervised ResNet-50 and Swin-T are pre-trained on ImageNet-1K, on which EsViT with Swin-T is pre-trained as well (Batch Size=512). EsViT outperforms its supervised counterpart, and is on par with the performance of CLIP in a similar image encoder architecture scale.

## C.6 PRE-TRAINING DATASETS

We describe the statistics and training schedule on larger and less curated datasets in Table 13. The pre-training epochs are chosen so that the model is trained with a similar number of augmented views.

| Name | Description | Size (#Images) | Epochs | Warmup |
|---|---|---|---|---|
| ImageNet-1K (Deng et al., 2009) | Images evenly distributed in 1K object concepts | 1.2 million | 300 | 10 |
| WebVision-v1 (Li et al., 2017) | Web images with 1K concept queries from ImageNet-1K | 2.4 million | 150 | 5 |
| OpenImages-v4 (Kuznetsova et al., 2020) | Diverse/complex scenes with several objects for detection | 7.5 million | 50 | 2 |
| ImageNet-22K (Deng et al., 2009) | Images distributed in 22K object concepts in a hierarchy | 14.2 million | 30 | 1 |

Table 13: Pre-train dataset statistics and training schedule.

## C.7 RESULTS ON CORRESPONDENCE LEARNING

We first quantitatively evaluate the correspondence learning results with 50K images in the ImageNet validation dataset. We create a simple evaluation dataset with mild augmentations. For a center-crop image, we apply `HorizontalFlip`, then `ColorJitter` and `RandomGrayscale` to create a new augmented view. In this way, ground-truth correspondences are created. Please see the 1st row of Figure 9 for one such example. The top-10 correspondences are used for evaluation. Two metrics are considered: (1) Accuracy measures the percentage of correctly matched region pairs, (2) distance error indicates the averaged $\ell_2$ distance between the predicted matched region and ground-truth region (the value is 0 for perfect matching). The results are reported in Figure 8. DINO with monolithic Transformers shows surprisingly good performance on correspondence learning. The use of multi-stage Transformer architecture reduces this ability, shows a lack of good region correspondence. With $\mathcal{L}_R$, the region matching ability is significantly recovered.

In Figure 9, we visualize the correspondences for more images. Overall, DINO with monolithic Transformers is able to discover most salient correspondences of semantic meaning in the mild augmentation conditions, even without an implicit region matching loss in training. We believe this previously underestimated property is whole-noting, and has potentials to enable more applications. However, this desired property gets dilated when changing from monolithic to multi-stage Transformer architecture (from column 1 to column 2), then the proposed region level task can alleviate this issue (from column 2 to column 3).

To more specifically analyze the correspondences, we note the following results. The first row shows a simple case, where only images of left-to-right flipped views are presented. The ground-truth

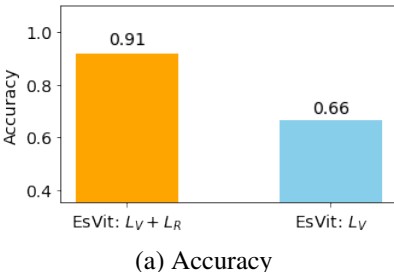
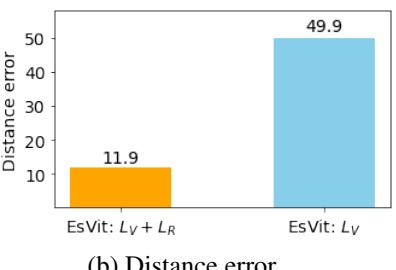

(a) Accuracy                    (b) Distance error

Figure 8: Quantitative evaluation on correspondence learning on ImageNet validation set. $\mathcal{L}_R$ can significantly improve correspondence learning quality for multi-stage architectures. As a reference, DINO ($\mathcal{L}_V$ with monolithic Transformer architecture) achieves 0.95 accuracy and 2.49 distance error, which we believe is a strong evidence to identify the intriguing property of automatic correspondence learning.

correspondences should be horizontal lines that link the two flipped regions. It reveals that the view-level pre-train task alone is insufficient to learn good correspondences for the multi-stage Transformer architecture, while region matching task can alleviate this issue significantly. Similar observations are shown in row 3 and row 4.

We further study more cases that requires real-world correspondences in row 2, row 5 and row 6. These views are not generated with data augmentation (as in model pre-training), but are often presented in more practical scenarios: one-to-many mappings, cartoon-to-toy, seasonal changing of the scene, respectively. The proposed region matching task can work particularly well in those cases.

### C.8 MORE VISUALIZATION RESULTS OF ATTENTION MAPS

We visualize attention maps at the top layer in Figure 10, 11, 12. With a monolithic Transformer architecture, DINO can automatically identify the main foreground objects. Unfortunately, changing from monolithic to the multi-stage Transformer architecture (From left column to middle column), this property gets lost. There are more heads in the multi-stage architecture than monolithic architecture (24 heads vs 6 heads in this case) in the last year. A fair number of heads in EsViT shows redundant patterns, this issue can be reduced when the region-level matching task is added (From middle column to right column).

We observed that DINO with monolithic Transformer architecture only learns to attend the foreground objects, even when the query is a background region (see Figure 12). This is perhaps because DINO models are trained to learn view-level invariance, the main objects in the pre-train dataset ImageNet tend to be the principle factor that remains invariant across different augmented views. Hence, all backgrounds are ignored, regardless of the query positions. This is improved in EsViT with the region-level pre-train task, as the model is trained to match individual regions.

DINO shows high entropy values in all of 6 heads (perhaps a required condition to cover all regions of the main object). In EsViT, $\mathcal{L}_R$ plays an interesting role in modulating the entropy distributions among heads: it increases those with larger entropy values, while decreasing those with lower entropy values. In another word, it makes the attention patterns in different heads more diverse.

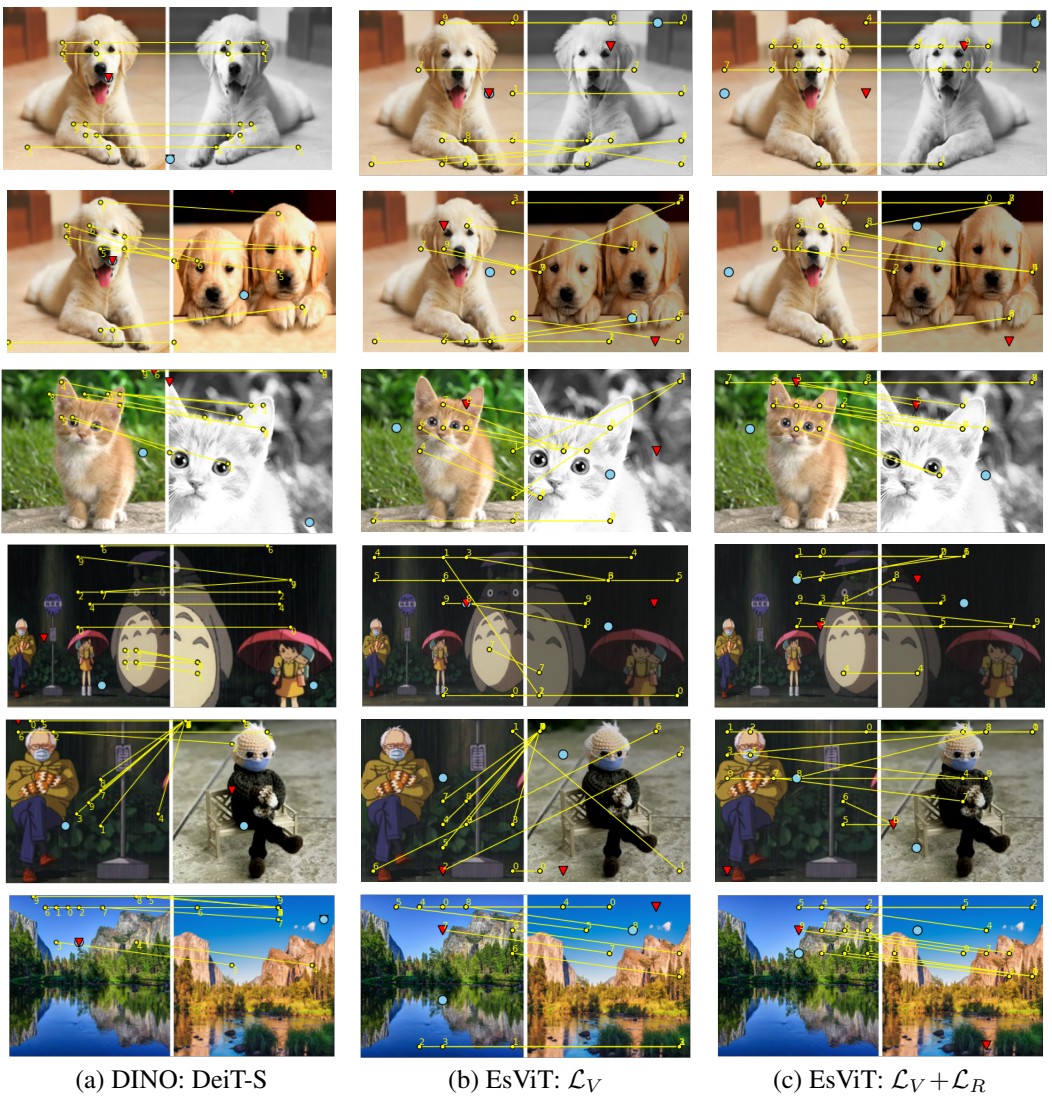

| (a) DINO: DeiT-S | (b) EsViT: $\mathcal{L}_V$ | (c) EsViT: $\mathcal{L}_V + \mathcal{L}_R$ |
| --- | --- | --- |

Figure 9: The learned correspondences. Yellow lines are the top-10 correspondences between two views, where the numbers indicates the rankings of similarity scores, yellow dots with the same number are paired. The blue dot and red triangle indicates the most similar local regions that correspond to the global feature of the view itself and the other view, respectively. Please zoom in for detailed correspondence mappings.

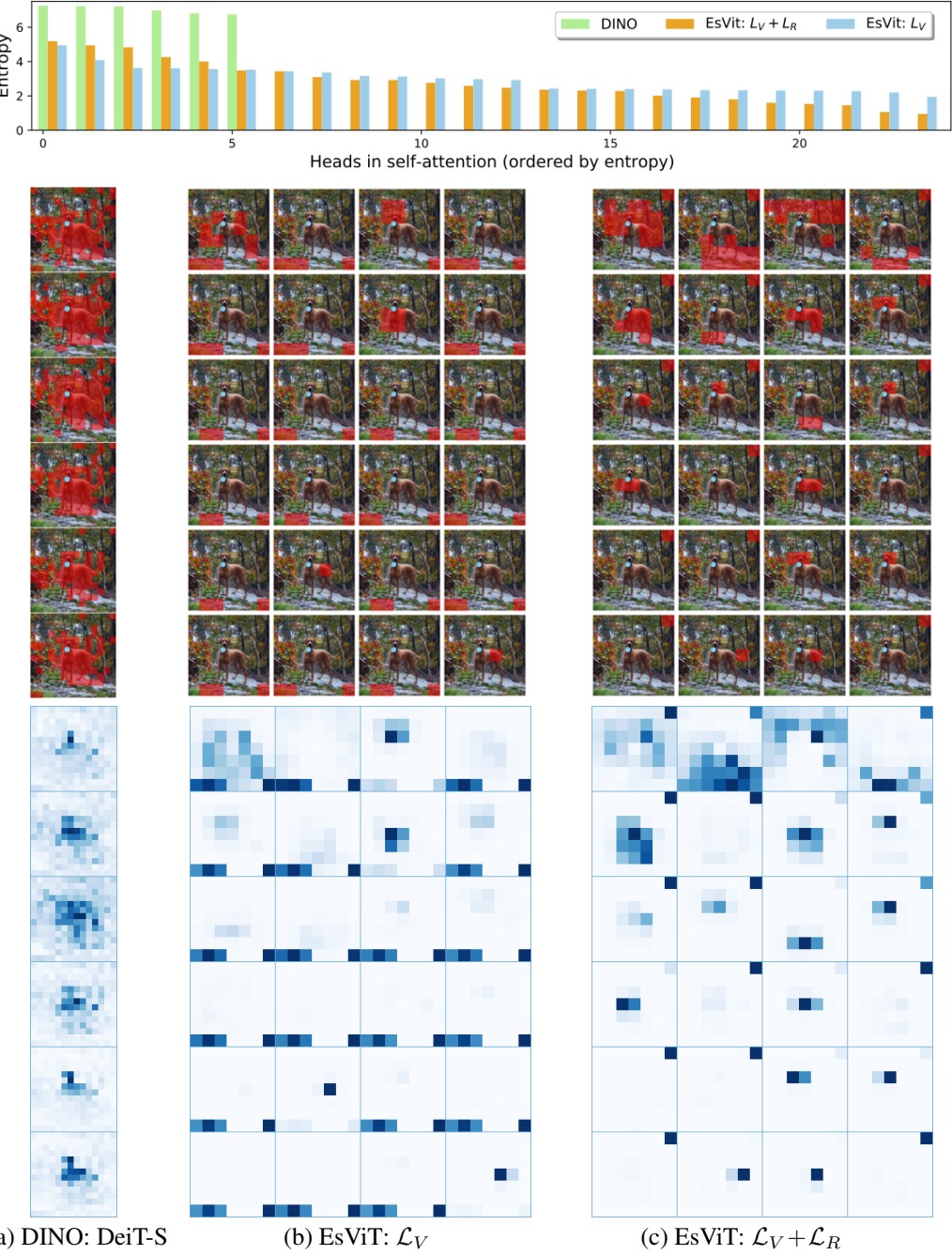

(a) DINO: DeiT-S     (b) EsViT: $\mathcal{L}_V$     (c) EsViT: $\mathcal{L}_V + \mathcal{L}_R$

Figure 10: The learned attention maps for all heads at the top layer, ranked by the entropy of softmax probability. Query is the blue dot in the top-left of the image. Top: Entropy of each heads. Middle: top 60% probability mass. Bottom: full attention maps. $\mathcal{L}_R$ shows more attention patterns than $\mathcal{L}_V$ only.

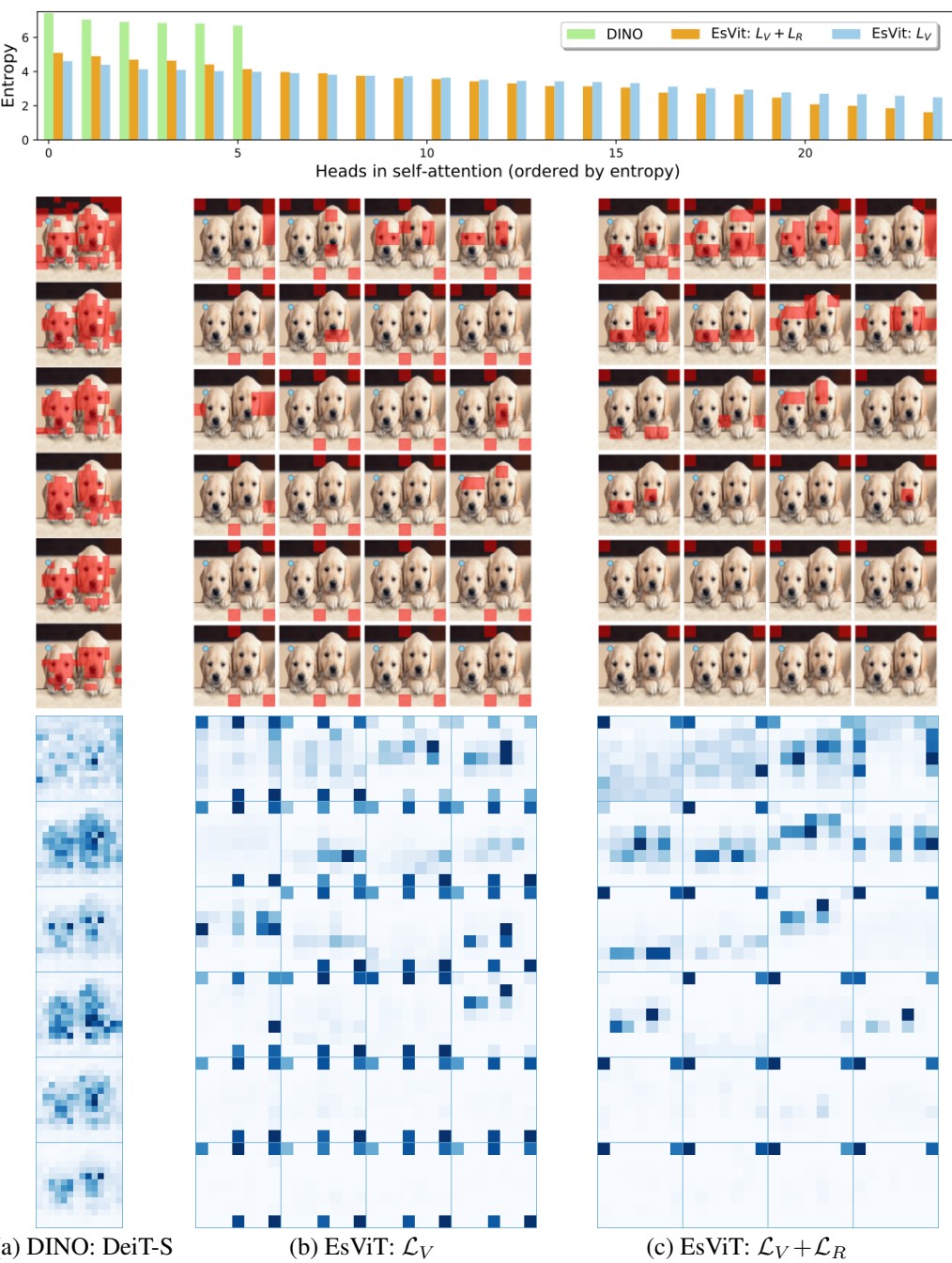

Figure 11: The learned attention maps for all heads at the top layer, ranked by the entropy of softmax probability. Query is the blue dot in the center of the image. Top: Entropy of each heads. Middle: top 60% probability mass. Bottom: full attention maps. $\mathcal{L}_R$ shows more attention patterns than $\mathcal{L}_V$ only.

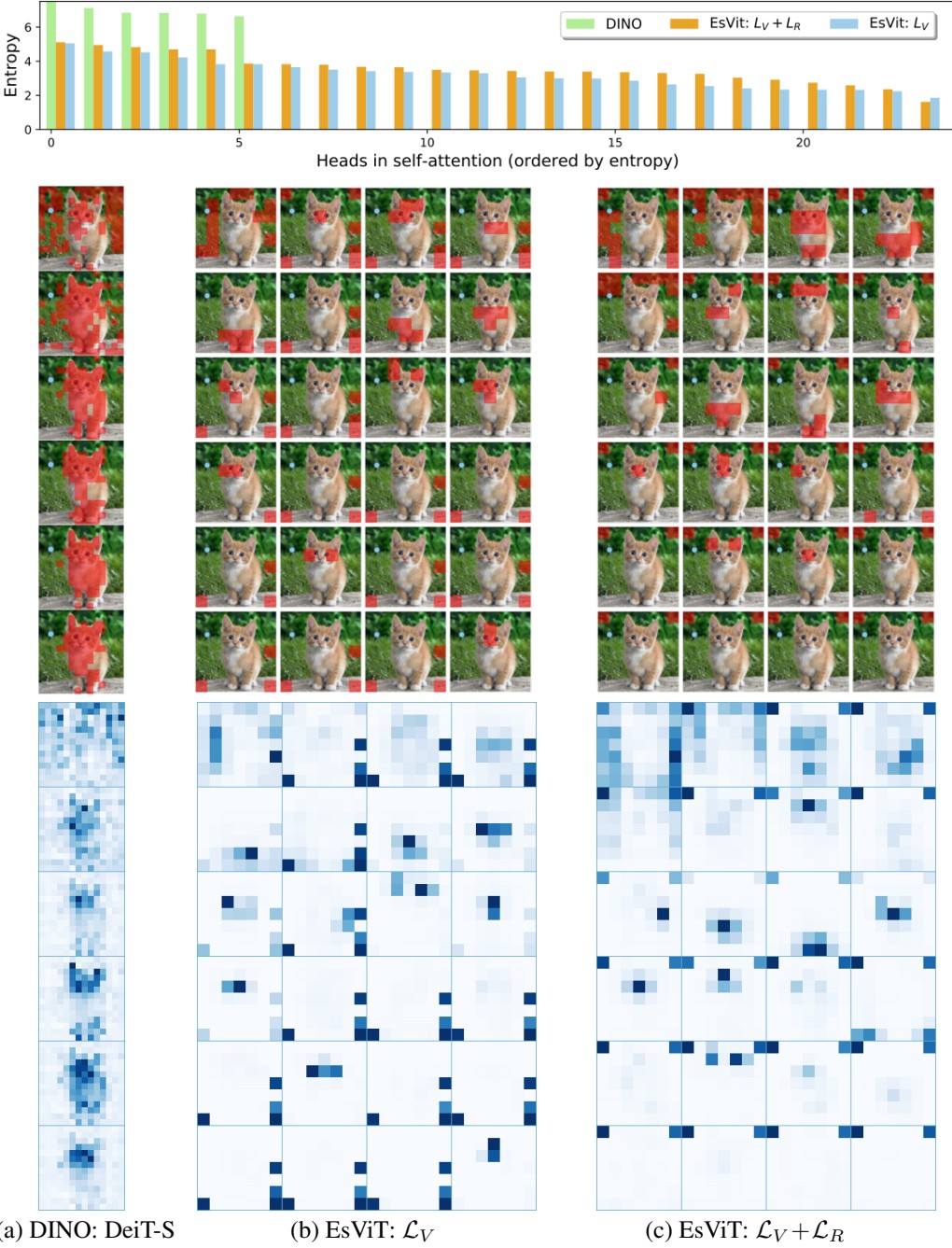

Figure 12: The learned attention maps for all heads at the top layer, ranked by the entropy of softmax probability. Query is the blue dot in the top-left of the image. Top: Entropy of each heads. Middle: top 60% probability mass. Bottom: full attention maps. DINO mainly attends the main object even when the query is a background region.

