# OpenReview forum: "Efficient Self-supervised Vision Transformers for Representation Learning"
_ICLR.cc/2022/Conference — ICLR 2022 Poster_

### Official Review · Reviewer_nCrU · 2021-11-01

**Correctness:** 4
**Technical Novelty And Significance:** 3
**Empirical Novelty And Significance:** 3
**Recommendation:** 8
**Confidence:** 4

**Main Review:**

Weaknesses:
1) I find the contribution, in the form of the loss term, incremental with respect to Swin, and the performance gain, demonstrated in most experiments, modest.
2) In my opinion, Figure 1 and Table 1 misrepresent the performance gain that stems from the contribution of the paper. The baseline architecture (Swin-B and Swin-B/W=14) already outperforms almost all methods in the comparison at a significantly lower number of parameters and much higher throughput. (The corresponding numbers are reported in Tab 5.) \
I encourage the authors to modify Figure 1 and Table 1 to represent the increase in performance of Swin resulting from the use of the proposed loss term. This can be done by introducing to both parts of Figure 1 points representing the performance of the base architectures (Swin-B, -S, and -T), trained without $\mathcal{L}_R$. The performance gain due to training them with $\mathcal{L}_R$ can be indicated in the plot with vertical arrows. Table 1 can be modified by reporting the performance of the base architectures trained without the proposed loss, in addition to their performance when trained with this loss. (Basically, by merging Tab 5 into Tab 1. The "ablation" results can still be discussed in a separate section.)
3) In the current transfer learning results, the impact of the authors' contribution on transferability of the trained networks is not clear. I would prefer the experiments in learning to classify 18 small datasets to highlight the performance gain/loss that stems from training with $\mathcal{L}_R$, in addition to the current comparison of supervised to unsupervised training. This would imply adding a third bar color to Figure 3 and one more row to both parts of Table 2, to represent the performance of Swin trained in a self-supervised regime, but without $\mathcal{L}_R$.

Strengths:
* A. I like the story-line of the paper, where an interesting observation about the discriminative (matching) power of patch descriptors motivates the construction of the loss term.
* B. The proposed loss term is shown to improve both the performance of Swin and of other architectures.
* C. The experimental evaluation is extensive, in that it also shows limitations of the proposed loss, for example, when applied to image segmentation (Tab 2 and Tab 6).
* D. According to me, the manuscript is well written and easy to understand.

Minor questions and editorial suggestions:
* Q1: in the loss term (2), you select $j^*$ by the cosine distance, but you minimize cross entropy. Have you tried using the same score for both tasks, i.e., selecting $j^*$ with the cross-entropy and minimizing cross entropy, or selecting $j^*$ by the cosine distance and minimizing the cosine distance?
* Q2: in the loss term (2), why haven't you tried optimal 1-1 matching (the Hungarian algorithm) to match source and target patches?
* Q3: in the paragraph "design choices of $\mathcal{L}_R$", you describe the choice of "argmax" vs "Optimal Transport" in the loss term (2). I do not understand how OT can be used to select the optimal index $j^*$. Could you clarify this?
* ES1. page 2: "loss of this property", "to alleviate this issue": Perhaps use enumeration instead of bullets and make the statements more specific: loss of the property described in 1, alleviate the issue identified in point 2.
* ES2. page 3: "ViT yields 95\% accuracy in (...) region-to-region correspondences" - consider specifying how you compute the correspondence scores/patch distances. (cosine similarity, as in subsec 4.4 page 8 ?)
* ES3. in the caption of Tab. 1, please specify W=14 is the window size.

**Summary Of The Paper:**

The contribution of the paper comprises:
* the observation that the Multi-Stage Vision Transformer (MSVT), as opposed to the "monolithic" Vision Transformer (VT), does not produce discriminative patch representation, and
* a loss function for self-supervised pre-training of MSVTs, that encourages discriminative patch representation.

Extensive experimental evaluation demonstrates that the proposed loss term, when added to the standard non-contrastive, self-supervised loss, brings a modest but systematic improvement in performance of the MSVT (Tab 5) and other architectures (Tab 4).


**Summary Of The Review:**

I find the main idea of this paper convincing, the manuscript well written, and the evaluation thorough. The fact that the authors highlight some limitations of their loss term, in particular when applied to image segmentation, attests to their scientific scrutiny. I think the paper deserves a publication. I am willing to rate it 8/10, instead of 10/10, because I find the contribution incremental and the performance gain modest.

In this round of reviews, I lower my recommendation to 6, because I find the weakness (2), described above, acute. It misrepresents the effect of the contribution on performance in the teaser figure and in the first table with quantitative results. I am confident the authors can fix it in the next revision of the paper, which will prompt me to increase the rating to 8/10.

I encourage the authors to also address the weakness (3) in the next revision of the paper. The current transfer learning experiments (section 4.2) do not evaluate the specific contribution of the paper, but rather answer more general questions. I am confident that adding evaluation more focused on the use of $\mathcal{L}_R$ will vastly benefit this part of the paper. I nevertheless do not feel that I should request completing this task within the limited review period, and I will not lower my recommendation on this basis.

EDIT:
the authors addressed the weaknesses 2) and 3) listed in the main review and answered all my questions. I am therefore happy to endorse the paper. This is a solid work and deserves to be published.

---

> ### Author Response · Authors · 2021-11-21
> **Response to Reviewer nCrU**
>
> > In my opinion, Figure 1 and Table 1 misrepresent the performance gain that stems from the contribution of the paper.
>
> Figure 1 and Table 1 have been updated to add the suggested model variant trained with $\mathcal{L}_R$ only. We have removed Table 5, and merged related numbers in Table 1.
>
> ----
>
> > I would prefer the experiments in learning to classify 18 small datasets to highlight the performance gain/loss that stems from training with L_R, in addition to the current comparison of supervised to unsupervised training. This would imply adding a third bar color to Figure 3 and one more row to both parts of Table 2.
>
> Figure 3 and Table 2 have been updated to add the suggested model variant trained with $\mathcal{L}_R$ only.
>
> ----
> > in the loss term (2), you select  by the cosine distance, but you minimize cross entropy. Have you tried using the same score for both tasks, i.e., selecting  with the cross-entropy and minimizing cross entropy, or selecting  by the cosine distance and minimizing the cosine distance?
>
> No, we have not tried using the same scores for simplicity. Our initial show that cross-entropy loss is preferred to compare $p$, which involves a temperature hyper-parameter.  Applying cross-entropy loss for $z$ requires introducing and tuning an additional temperature hyper-parameter.
>
> In our experiments, we always choose to select the best matching index using cosine similarity and update model weights using cross-entropy. It is preferred to use cross-entropy loss than MSE loss to update the model, as shown in our experiments (see the paragraph of "Design choices of $\mathcal{L}_R$"). Since $cosine$-$distance(a,b)$ is equivalent to MSE$(a,b)$ when $a$ and $b$ are $L2$-normalized vectors, so we would suggest to use cross-entropy.
>
> ----
>
> > In the loss term (2), why haven't you tried optimal 1-1 matching (the Hungarian algorithm) to match source and target patches?
>
> Thanks for the suggestions, we did not try the Hungarian algorithm. Both Hungarian and OT algorithms aim at obtaining better global alignment/matching results between source and target patches. We chose to experiment with OT, as it has shown better results than Hungarian in other application domains such as object detection [*].
>
>  [*] OTA: Optimal Transport Assignment for Object Detection, CVPR 2021
>
> ----
> > In the paragraph "design choices of $\mathcal{L}_R$", you describe the choice of "argmax" vs "Optimal Transport" in the loss term (2). I do not understand how OT can be used to select the optimal index . Could you clarify this?
>
> The original cosine similarity matrix on $z$ is used as input for the OT, which outputs a transportation plan (and an updated similarity matrix) at a minimal cost. The arg max is applied to the new similarity matrix to obtain the assignment.
>
> ----
> > ES2. page 3: "ViT yields 95% accuracy in (...) region-to-region correspondences" - consider specifying how you compute the correspondence scores/patch distances. (cosine similarity, as in subsec 4.4 page 8 ?)
>
> Yes. The cosine similarity is used to compute correspondences,  which uses the same process with subsec 4.4 on page 8.
>
> Thanks for the detailed suggestions on ES1. and ES3 as well. We have revised the paper accordingly.

---

> > ### Comment · Reviewer_nCrU · 2021-11-21
> > **Thank you; I updated my recommendation.**
> >
> > The changes to the manuscript address all my concerns.
> > Thank you also for answering my questions.
> > I have updated my recommendation to 8/10.
> > I think this is a solid work that deserves to be published.

---

### Official Review · Reviewer_6vXm · 2021-11-02

**Correctness:** 4
**Technical Novelty And Significance:** 2
**Empirical Novelty And Significance:** 3
**Recommendation:** 6
**Confidence:** 5

**Main Review:**

Strengths:
(1) It introduces the new pretraining task to capture fine-grained region dependencies;
(2) The experimental results are good and better than other compared approaches.

Weaknesses:
(1) The patch merging module and sparse self-attention in the multi-stage ViT are very similar to the patch merging in the paper Swin Transformer [Liu et al 2021]. The authors should clearly explain the differences between this paper and Swin Transformer.
(2)The equation (2) should add more high-level descriptions. For each local feature z_i, why finding the local feature z_j from the teacher with the highest cosine similarity can capture the fine-grained region dependency?

**Summary Of The Paper:**

This paper develops an efficient self-supervised vision transformer for learning visual representations. It introduces a multi-stage architecture with sparse attentions to reduce computation complexity and proposes a new pretraining task of region matching to capture fine-grained region dependencies. The results on the ImageNet and 18 small datasets or downstream tasks are good and compared with other state-of-the-art approaches.


**Summary Of The Review:**

The theoretical novelties are limited, this paper mainly borrows the ideas from two papers DINO [Caron et al. 2021] and Swin transformer [Liu et al 2021]. However, the experimental results are very good in this paper, and the paper provides a lot of implementation details for others to reproduce the results.

---

> ### Author Response · Authors · 2021-11-21
> **Response to Reviewer 6vXm**
>
> > The patch merging module and sparse self-attention in the multi-stage ViT are very similar to the patch merging in the paper Swin Transformer [Liu et al 2021]. The authors should clearly explain the differences between this paper and Swin Transformer.
>
> When the Swin Transformer is employed as the backbone, the merging module and sparse self-attention component is the same as the original paper. The only minor difference is that we add special treatments to deal with input augmented views of different resolutions, when the resolution (feature map size more specifically) is not divisible by the window size.
>
> We clarify that novel multi-stage ViT architecture itself is NOT the major contribution of this paper.  We focus on how to best use multi-stage ViT in the self-supervised settings, by (as summarized by Reviewer nCrU):
> - Identifying an intriguing property of monolithic ViT, and raising the issue that the multi-stage ViT loses this property
> - A region-matching loss to alleviate the above issue to improve learned visual representations.
>
> ----
>
> > The equation (2) should add more high-level descriptions. For each local feature $z_i$, why finding the local feature $z_j$ from the teacher with the highest cosine similarity can capture the fine-grained region dependency?
>
> Note that $i$ and $j$ are the indexes of best matched two regions (measured by cosine similarity). $z_i$ and $z_j$ are **contextualized** features, in which the neighboring region information from two different augmented views is also encoded. This contextualized information is also included in the probability outputs: $p_i = h_0(z_i)$ and $p_j = h_0(z_j)$. Minimizing the cross-entropy between $p_i$ and $p_j$ encourages different contexts (surrounding regions) of the best matched two regions to learn invariant features, and thus captures the region-dependency.
>
> We have added more high-level descriptions for equation (2) in the revision.

---

### Official Review · Reviewer_sJFu · 2021-11-05

**Correctness:** 4
**Technical Novelty And Significance:** 3
**Empirical Novelty And Significance:** 3
**Recommendation:** 8
**Confidence:** 4

**Main Review:**

The paper has many strengths.

- It is interesting to see that the loss on local features can work well in the self-supervised learning case. This observation may find more usage in the future works of doing self-supervised learning on transformer-based models.

- The experiments are comprehensive and convincing. The paper tries the same idea on multiple transformer structures and different tasks. The proposed method can achieve the efficiency of the multi-stage transformer models while achieving good self-supervised feature pre-training for tasks including classification, segmentation, and detection.

- The info in the appendix is helpful for result reproduction and detailed understanding.

In terms of weakness, I do hope that the paper can be more clearly written, such as reorganizing the info between main text and appendix to give more intuition of L_R.

In addition, the detail of Table 2 is too scarce to understand directly. I guess the supervised baseline is Swin-T, but it is hard to infer from the paper directly.

**Summary Of The Paper:**

The paper investigates how to use self-supervised learning for multi-stage visual transformer models. Previous works have shown that SSL can learn image correspondences and lead to performant pre-trained models, while the multi-stage models can reduce the computation cost dramatically. This work tries to merge these two trends together. The solution is a new region-based loss that can be applied to the local features. The comprehensive experiments show the advantages of the resulting models on multiple tasks.

**Summary Of The Review:**

Overall, the manuscript looks like a good paper. I do hope reading the paper can be easier despite the page limit.

---

> ### Author Response · Authors · 2021-11-21
> **Response to Reviewer sJFu**
>
> > I do hope that the paper can be more clearly written, such as reorganizing the info between main text and appendix to give more intuition of $\mathcal{L}_R$.
>
> Thanks for the suggestions. We tried again to move more information on $\mathcal{L}_R$ from appendix to the main text, including:
> - In Section 2.1,  we add the experimental process to quantify the lost property of correspondence learning, and thus to motivate $\mathcal{L}_R$.
> - In Section 2.2, we added more intuitive explanations on $\mathcal{L}_R$.
>
> ----
>
> >  The detail of Table 2 is too scarce to understand directly. I guess the supervised baseline is Swin-T, but it is hard to infer from the paper directly.
>
> The reviewer is right that the supervised baseline is Swin-T trained with 3x schedule. We have added the information in the revision.

---

### Author Response · Authors · 2021-11-21
**Summary of revision**

We thank all reviewers for their valuable feedback.  We have carefully revised our paper based on the suggestions from reviewers, with the main changes marked in blue. Below we provide our point-to-point response to each reviewer.

---

### Decision · Program_Chairs · 2022-01-20

**Decision:**

Accept (Poster)

**Comment:**

This paper proposes two techniques for improving self-supervised learning with a vision transformer. The first improvement is using a multi-stage ViT, which is very similar to Swin transformer and authors recognized this is not a major contribution. The authors further found that using a multi-stage ViT does not produce discriminative patch representation, thus proposing the second improvement with a region level loss. While both improvements are not particularly novel by themselves, combining both leads to a strong empirical result. However, It does looks like the multi-scale vision transformer is the major improvement as removing the regional loss only leads to less than 1% decrease in performance in most cases. In general this is a good "engineering" paper with a practical approach for improving self-supervised learning with vision transformation and obtained strong results, thus it's worthy of publication.